# Genome-wide association meta-analysis of spontaneous coronary artery dissection identifies risk variants and genes related to artery integrity and tissue-mediated coagulation

Spontaneous coronary artery dissection (SCAD) is an understudied cause of myocardial infarction primarily affecting women. It is not known to what extent SCAD is genetically distinct from other cardiovascular diseases, including atherosclerotic coronary artery disease (CAD). Here we present a genome-wide association meta-analysis (1,917 cases and 9,292 controls) identifying 16 risk loci for SCAD. Integrative functional annotations prioritized genes that are likely to be regulated in vascular smooth muscle cells and artery fibroblasts and implicated in extracellular matrix biology. One locus containing the tissue factor gene *F3*, which is involved in blood coagulation cascade initiation, appears to be specific for SCAD risk. Several associated variants have diametrically opposite associations with CAD, suggesting that shared biological processes contribute to both diseases, but through different mechanisms. We also infer a causal role for high blood pressure in SCAD. Our findings provide novel pathophysiological insights involving arterial integrity and tissue-mediated coagulation in SCAD and set the stage for future specific therapeutics and preventions.

Cardiovascular disease is the leading cause of death in women, but sex-specific aspects of the risk of heart disease and acute myocardial infarction (AMI) remain understudied[1]. Spontaneous coronary artery dissection (SCAD) and atherosclerotic coronary artery disease (CAD) are both causes of acute coronary syndromes leading to AMI[2–6]. However, in contrast with CAD, SCAD affects a younger, predominantly female population[7] and arises from the development of a hematoma, leading to dissection of the coronary tunica media with the eventual formation of a false lumen, rather than atherosclerotic plaque erosion or rupture[8]. SCAD has been clinically associated with migraine[9] and extra-coronary arteriopathies, including fibromuscular dysplasia (FMD)[10–13]. However, co-existent coronary atherosclerosis is uncommon[8,14]. While the genetic basis of CAD is increasingly well established[15], the pathophysiology of SCAD remains poorly understood[4]. The search for highly penetrant mutations in candidate pathways or by sequencing has garnered a low yield, often pointing to genes involved in other clinically undiagnosed inherited syndromes manifesting as SCAD[16]. Previous investigations of the impact of common genetic variation on the risk of SCAD have described five confirmed risk loci[17–20].

In this Article, we performed a meta-analysis of genome-wide association studies (GWASs) comprising 1,917 SCAD cases and 9,292 controls of European ancestry. We identified 16 risk loci, including 11 new association signals, demonstrating a substantial polygenic heritability for this disease. Importantly, we show that several common genetic risk loci for SCAD are shared with CAD but have a directionally opposite effect and a different genetic contribution of established

e-mail: da134@leicester.ac.uk; nabila.bouatia-naji@inserm.fr

**Table 1 | Lead associated variants at genome-wide significance in SCAD loci**

| Locus | Chr:position | rsID | Annotated gene(s) | EA | OA | EAF | SCAD GWAS meta-analysis (1,917 cases and 9,792 controls) | | | |
| | | | | | | | OR (95% CI) | $P^a$ | Direction[b] | Het[c] |
|---|---|---|---|---|---|---|---|---|---|---|
| 1 | 1:59656909 | rs34370185 | FGGY-DT | T | G | 0.29 | 1.34 (1.24–1.46) | $1.4×10^{-12}$ | ++++++++ | 0.04 |
| 2 | 1:95050472 | rs1146473 | F3 | C | T | 0.19 | 1.32 (1.20–1.45) | $5.8×10^{-9}$ | ++++++++ | 0.10 |
| 3[d] | 1:150504062 | rs4970935 | ECM1/ADAMTSL4 | C | T | 0.28 | 1.72 (1.59–1.87) | $6.1×10^{-39}$ | ++++++++ | 0.64 |
| 4 | 4:7774352 | rs6828005 | AFAP1 | G | A | 0.45 | 1.29 (1.20–1.40) | $2.6×10^{-11}$ | ++++++++ | 0.82 |
| 5 | 4:146788035 | rs1507928 | ZNF827 | C | T | 0.48 | 1.25 (1.16–1.35) | $8.9×10^{-9}$ | ++++++++ | 0.38 |
| 6 | 5:52155642 | rs73102285 | ITGA1 | G | A | 0.27 | 1.27 (1.17–1.38) | $1.1×10^{-8}$ | ++++++− | 0.31 |
| 7[d] | 6:12903957 | rs9349379 | PHACTR1 | A | G | 0.62 | 1.64 (1.51–1.78) | $2.9×10^{-32}$ | ++++++++ | 0.19 |
| 8 | 10:124259062 | rs2736923 | HTRA1 | A | G | 0.89 | 1.44 (1.26–1.64) | $4.6×10^{-8}$ | ++++++?+ | 0.60 |
| 9 | 11:95308854 | rs11021221 | SESN3 | A | T | 0.17 | 1.47 (1.33–1.61) | $4.1×10^{-15}$ | ++++++++ | 0.19 |
| 10[d] | 12:57527283 | rs11172113 | LRP1 | T | C | 0.62 | 1.62 (1.49–1.76) | $9.0×10^{-31}$ | ++++++++ | 0.70 |
| 11 | 12:89978233 | rs1689040 | ATP2B1 | C | T | 0.59 | 1.28 (1.18–1.39) | $7.0×10^{-10}$ | +++++++− | 0.66 |
| 12 | 13:110838236 | rs7326444 | COL4A1 | G | A | 0.64 | 1.31 (1.21–1.42) | $1.0×10^{-10}$ | ++++++++ | 0.52 |
| 12 | 13:111040681 | rs11838776 | COL4A2 | G | A | 0.73 | 1.50 (1.37–1.65) | $2.5×10^{-18}$ | +++++++− | 0.42 |
| 13[d] | 15:48763754 | rs7174973 | FBN1 | G | A | 0.11 | 1.54 (1.37–1.72) | $1.6×10^{-13}$ | ++++++++ | 0.03 |
| 14 | 15:71628370 | rs10851839 | THSD4 | A | T | 0.68 | 1.32 (1.22–1.44) | $5.5×10^{-11}$ | ++++++−+ | 0.24 |
| 15[d] | 21:35593827 | rs28451064 | MRPS6/SLC5A3/KCNE2 | G | A | 0.88 | 2.04 (1.77–2.35) | $1.2×10^{-22}$ | ++++++++ | 0.50 |
| 16 | 22:33282971 | rs137507 | TIMP3 | T | C | 0.11 | 1.38 (1.23–1.55) | $3.3×10^{-8}$ | +++++−++ | 0.02 |

[a]Unadjusted $P$ value of association obtained by two-sided Wald test. [b]Direction signs for the individual association results in the DISCO-3C, SCAD-UK I, Mayo Clinic, CanSCAD/MGI, VCCRI I, SCAD-UK II, VCCRI II and DEFINE-SCAD studies, respectively. [c]$P$ values from the Cochran's Q statistic heterogeneity test. [d]Loci previously reported in SCAD. EA, effect allele; EAF, effect allele frequency; OA, other allele; OR, odds ratio.

cardiovascular risk factors. These findings implicate arterial integrity related to extracellular matrix biology, vascular tone and tissue coagulation in the pathophysiology of SCAD.

## Results

### GWAS meta-analysis and single-nucleotide polymorphism heritability

We conducted a GWAS meta-analysis of eight independent case–control studies (Supplementary Figs. 1 and 2 and Supplementary Table 1). Sixteen loci demonstrated genome-wide-significant signals of association with SCAD, among which 11 were newly described for this disease (Table 1, Fig. 1a, Supplementary Table 2 and Supplementary Fig. 3). One locus on chromosome 4 (AFAP1) was recently reported for SCAD in the context of pregnancy[19] and has now been confirmed as being generally involved in SCAD (Table 1). The estimated odds ratios of associated loci ranged from 1.25 (95% confidence interval (CI) = 1.16–1.35) in ZNF827 on chromosome 4 to 2.04 (95% CI = 1.77–2.35) on chromosome 21 near KCNE2 (Table 1). We report evidence for substantial polygenicity for SCAD with an estimated single-nucleotide polymorphism (SNP)-based heritability above 0.70 ($h^2_{SNP} = 0.71 ± 0.11$ on the liability scale using linkage disequilibrium score regression[21] and $h^2_{SNP} = 0.70 ± 0.12$ using SumHer[22]; Supplementary Table 3). The ECM1/ADAMTSL4 locus on chromosome 1 accounted for the largest proportion of heritability for SCAD in our dataset ($h^2 = 0.028$), followed by the COL4A1/COL4A2 locus, which contained two independent GWAS signals ($h^2 = 0.022$; Supplementary Table 4 and Supplementary Fig. 4). Overall, we estimate that the 16 loci explain ~24% of the total SNP-based heritability of SCAD (Supplementary Table 4).

### Functional annotation of variants in SCAD loci

We found SCAD-associated variants to be significantly enriched in enhancer marks specific to gene expression in arterial tissues from ENCODE[23] (for example, the aorta, tibial artery, thoracic aorta and coronary artery), as well as several tissues rich for smooth muscle cells

(for example, the colon, small intestine and uterus) (Supplementary Fig. 5). Based on recently published analyses of single-cell open chromatin in 30 adult tissues[24], we determined that vascular smooth muscle cells (VSMCs) and fibroblasts were the top enriched cell types for SCAD-associated loci among clusters represented in aorta and tibial artery datasets (Fig. 2a and Supplementary Fig. 6). Consistently, all but one SCAD locus included at least one variant that overlapped with enhancer marks or open chromatin peaks in coronary artery tissue, VSMCs or fibroblasts (Supplementary Fig. 7 and Supplementary Table 5). Among the top associated variants for SCAD, 14 were expression quantitative trait loci (eQTLs) for nearby genes in the aorta, coronary or tibial artery, whole blood or cultured fibroblasts (Fig. 1b and Supplementary Table 5).

### Tissue coagulation as a novel mechanism in SCAD

We applied a multi-source strategy to identify candidate genes located in risk or GWAS loci, or loci at risk for SCAD. We prioritized: (1) genes that were targets of eQTLs colocalizing with a GWAS signal (Supplementary Fig. 8a and Supplementary Table 6) or transcriptome-wide association study (TWAS) hits in at least one tissue relevant to arterial dissection (aorta, coronary or tibial artery, fibroblasts or whole blood from the Genotype Tissue Expression (GTEx) database) (Supplementary Fig. 8b and Supplementary Table 7); (2) genes with a biological function linked to the cardiovascular system in humans or mice; (3) genes involved in significant long-range chromatin conformation interactions from Hi-C data with SCAD-associated variants in the aorta[25]; and (4) those genes closest to or overlapping with the top associated variants. We identified one specific and strong candidate gene in 14 loci (Fig. 1b). For instance, the tissue factor gene F3 stood out as the most likely target gene near rs1146473 (odds ratio = 1.32; $P = 5.8 × 10^{-9}$)—a locus on chromosome 1 that we describe as novel for SCAD and any cardiovascular disease or trait so far. F3 is the closest coding gene to the association signal and was a TWAS hit in artery tissue (Supplementary Table 7). In addition, the rs1146473 risk allele for SCAD confidently

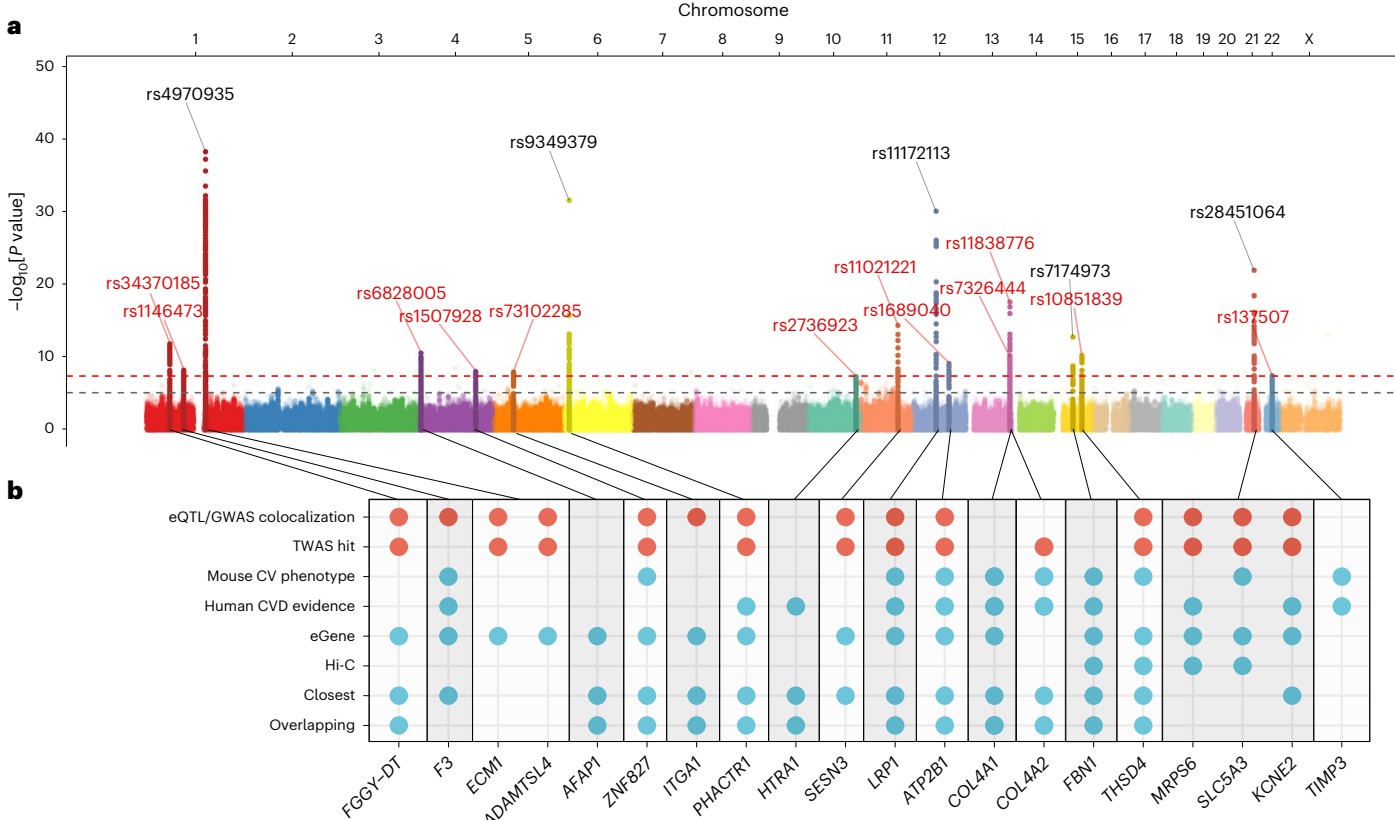

**Fig. 1 | GWAS meta-analysis main association results and gene prioritization at-risk loci. a**, Manhattan plot representation of SNP-based association meta-analysis in SCAD. The $x$ axis shows the genomic coordinates and the $y$ axis shows the $-\log_{10}[P\text{ value}]$ obtained by two-sided Wald test. SNPs located around genome-wide significant signals (±500 kb) are highlighted. The labels show the rsIDs for the lead SNPs, with newly identified loci in red and previously known loci in black. The dashed red line represents genome-wide significance ($P = 5 \times 10^{-8}$) and the gray line suggestive association ($P = 10^{-5}$). **b**, Summary of the strategy for the annotation of gene prioritization. The dots indicate genes fulfilling one of the following eight criteria: (1) colocalization of SCAD association signal and eQTL association in the aorta, coronary artery, tibial artery, fibroblasts or whole blood samples (GTEx version 8 release); (2) a TWAS hit in any of the above-mentioned tissues; (3) a cardiovascular (CV) phenotype in the gene knockout mouse; (4) existing evidence of gene function in cardiovascular disease (CVD) pathophysiology in humans; (5) the gene is an eGene for a nearby lead SNP in the above-mentioned GTEx tissues; (6) Hi-C evidence[25] for a promoter of the gene in a chromatin loop from human aorta tissue that includes variants from the credible set of causal variants; (7) the closest gene upstream or downstream from the lead SNP; or (8) variants in the credible set of causal variants map in the gene. Criteria 1 and 2 (blue dots) were given a tenfold weighted score over criteria 3–8. Genes with the most criteria were prioritized in each locus and are shown here.

(posterior probability = 94%) colocalized with an eQTL signal of *F3* in the aorta, supporting the genetic risk to potentially be the result of decreased *F3* expression in arteries (Fig. 2b and Supplementary Table 6). Tissue factor, also known as coagulation factor III, forms a complex with factor VIIa, which is the primary initiator of blood coagulation. Hence, reduced factor III expression is potentially a key biological mechanism contributing to hematoma formation in the coronary arteries of SCAD survivors. Consideration of genes encoding druggable targets, as derived by Finan et al.[26], indicated that tissue factor is a clinical phase drug candidate (tier 1 druggable target), with target reference numbers CHEMBL4081 (factor III) and CHEMBL2095194 (factor III/factor VII complex) (Supplementary Table 8).

To globally assess the biological mechanisms involving prioritized genes, we applied a network query based on Bayesian gene regulatory networks constructed from expression and genetics data from arterial tissues and fibroblasts[27–29]. We found extracellular matrix organization to be the biological function at which most prioritized genes and their respective immediate subnetworks clustered (Supplementary Fig. 9). Among the genes we prioritized in novel loci, a number encode proteins involved in extracellular matrix formation, including integrin alpha 1 (*ITGA1*), basement membrane constituent collagen type IV alpha

1 chain (*COL4A1*) and alpha 2 chain (*COL4A2*), serine protease HtrA serine peptidase 1 (*HTRA1*), metallopeptidase thrombospondin type 1 domain containing 4 (*THSD4*, encoding a partner of fibrillin 1, whose gene is located in a previously reported SCAD locus (*FBN1*)) and TIM metallopeptidase inhibitor 3 gene (*TIMP3*). Interestingly, integrin alpha 1, HTRA1 and collagen type IV subunits were labeled as potentially druggable targets based on their similarity to approved drug targets and members of key druggable gene families (tier 3; Supplementary Table 8). Of note, the *F3* subnetwork also clustered in extracellular matrix organization and connected with *HTRA1* and *TIMP3* subnetworks through Bayesian network edges from the aorta and coronary artery (Supplementary Fig. 9).

**Shared genetics between SCAD and arterial diseases**

With the exception of the *F3* locus, SCAD risk loci located within 1 megabase of the lead SCAD variants were at least suggestively ($P < 10^{-5}$) associated with other forms of cardiovascular and neurovascular disease. Using trait colocalization analyses, we found that the same variants were likely to be causal both for SCAD and the other diseases or traits at 15 loci (Fig. 3a and Supplementary Table 9). However, the directions of the effects were not systematically consistent across the loci for all of the diseases.

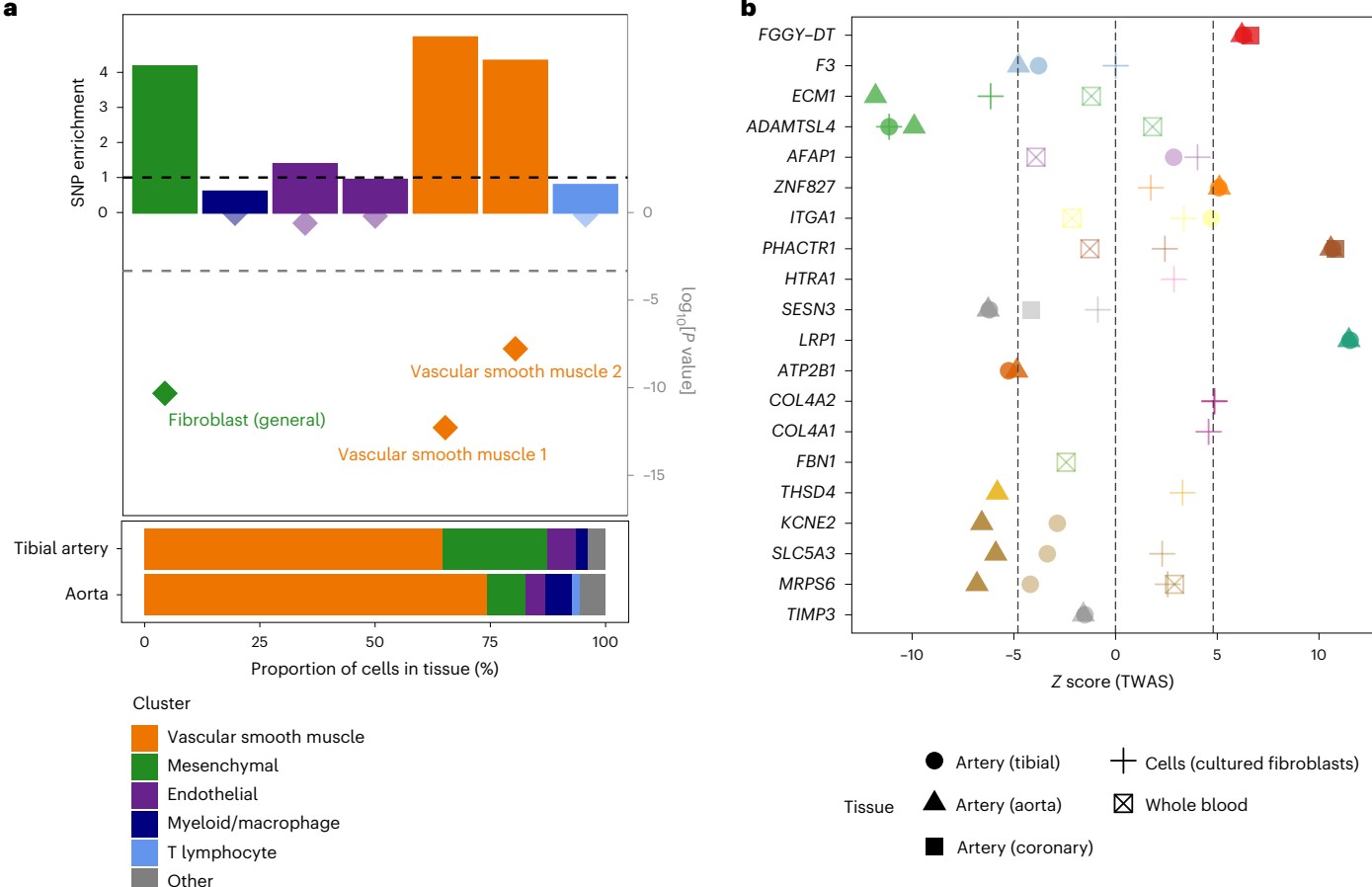

**Fig. 2 | Enrichment of SCAD SNPs in open chromatin regions from arterial cells and genetically predicted expression changes of nearby genes. a**, Top, representation of the fold-enrichment of SCAD SNPs (top *y* axis) and enrichment *P* value (log scale; bottom *y* axis) among the open chromatin regions of seven single-cell subclusters contributing to >1% of cells in artery tissue[24]. The SCAD 95% credible set of causal SNPs and their linkage disequilibrium proxies were matched to random pools of neighboring SNPs using the GREGOR package[43]. Enrichment represents the ratio of the number of SCAD SNPs overlapping open chromatin regions over the average number of matched SNPs overlapping the same regions. *P* values were evaluated by binomial one-sided test, with greater enrichment as the alternative hypothesis[43]. The bottom dashed line represents significance (*P* < 0.05) after adjustment for 105 subclusters. Higher opacity is used to identify significant associations (adjusted *P* < 0.05).

Bottom, composition of artery tissues relative to 105 single-cell subclusters, as determined by snATAC-seq in 30 adult tissues[24]. Only subclusters representing >1% of cells from either the aorta or tibial artery were represented. **b**, Representation of the SCAD TWAS *z* score for each prioritized gene in GWAS loci. The point shape indicates the tissue used in the TWAS association. The point color distinguishes genes located at different loci. The absence of a symbol indicates that the gene did not show significant heritability based on the eQTL data in the corresponding tissue. TWAS *P* values were calculated by two-tailed *z* test against a null distribution calculated by permutation for each gene or tissue[44]. Higher opacity is used to identify significant associations (Bonferroni adjusted *P* < 0.05), corresponding to a *z* score of >4.8 or <−4.8 (dashed gray lines).

---

Globally, SCAD loci showed evidence for high posterior probability for the same risk alleles to also probably be causal for FMD and cervical artery dissection (Fig. 3a and Supplementary Table 9). Linkage disequilibrium score regression-based genetic correlations indicated that SCAD correlates positively with FMD ($r_g = 0.38 \pm 0.18$; $P = 0.03$) and cervical artery dissection ($r_g = 0.61 \pm 0.20$; $P = 2.4 \times 10^{-3}$; Fig. 3b and Supplementary Table 10), which is consistent with the clinical observation of frequent coexistence of these arteriopathies in patients with SCAD. For instance, FMD is reported in ~40–60% of patients with SCAD[11,30]. Stratified analyses in the four largest case–control studies where FMD arteriopathies were screened indicated globally similar associations with SCAD (Supplementary Fig. 10 and Supplementary Table 11). Finally, genetic correlations indicated that SCAD positively correlates with several neurovascular diseases where predominantly arterial structure and/or function are altered, including stroke ($r_g = 0.17 \pm 0.06$; $P = 4.5 \times 10^{-3}$), migraine ($r_g = 0.18 \pm 0.06$; $P = 1.3 \times 10^{-3}$), intracranial aneurysm ($r_g = 0.22 \pm 0.06$; $P = 2.0 \times 10^{-4}$)

and subarachnoid hemorrhage ($r_g = 0.27 \pm 0.07$; $P = 6.4 \times 10^{-5}$) (Fig. 3b and Supplementary Table 10).

**Opposite genetic link between SCAD and CAD**

While patients with CAD are predominantly men (~75%) who often have pre-existing cardiometabolic comorbidities (mainly dyslipidemia, hypertension and type 2 diabetes), patients with SCAD are on average younger, present with fewer cardiovascular risk factors and are overwhelmingly women (>90%)[2,4]. Using genetic association colocalization and genetic correlation, we genetically compared SCAD with CAD. We found that, among SCAD loci, several were known to associate with CAD. Disease association colocalization analyses showed that for six loci SCAD and CAD are likely to share the same causal variants with high posterior probabilities (posterior probability of the shared causal variant hypothesis (H4) = 84–100%), but all with opposite risk alleles (Fig. 3a and Supplementary Table 7). Genetic correlation confirmed a genome-wide negative

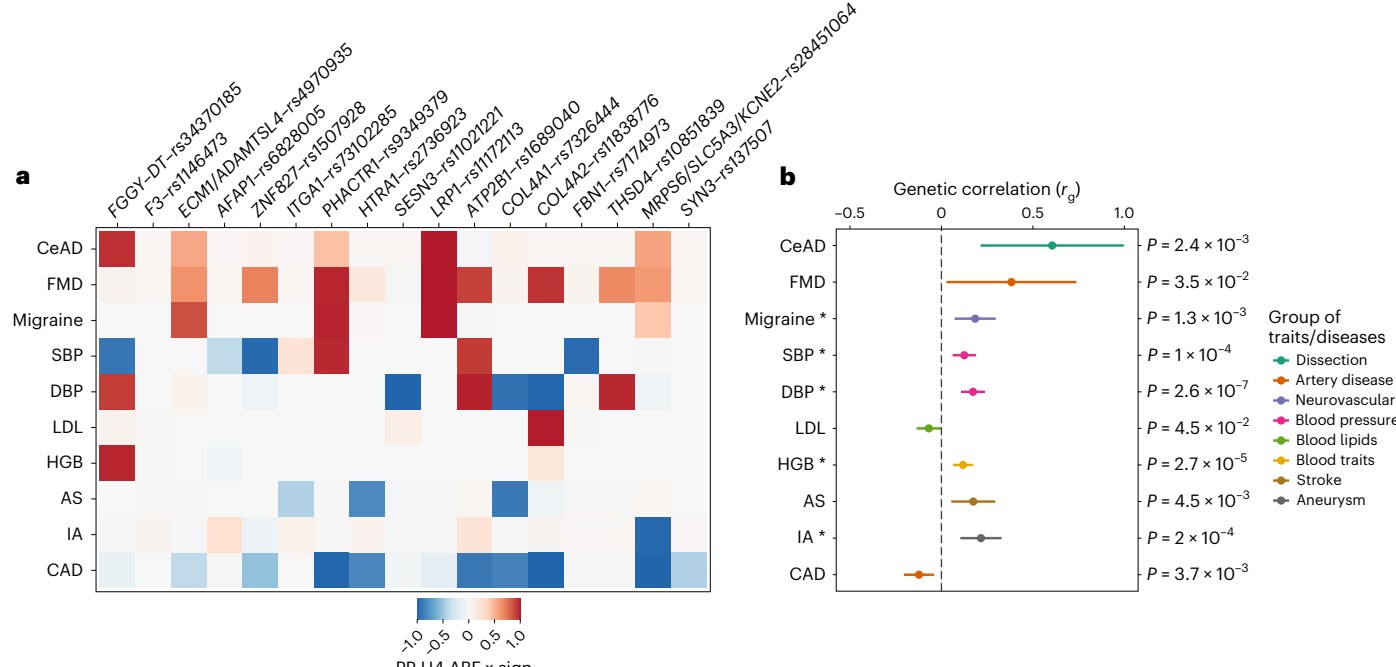

**Fig. 3 | Colocalization and genetic correlation of SCAD genetic association with cardiovascular diseases and traits. a,** Heatmap representing the colocalization of SCAD signals with GWAS analysis of the following cardiovascular diseases or traits: cervical artery dissection (CeAD), multifocal FMD, migraine, blood pressure (SBP and DBP), LDL cholesterol blood concentration, hemoglobin concentration (HGB), any stroke (AS), intracranial aneurysm (IA) and CAD. The tile color represents the H4 coefficient of approximate Bayes factor (ABF) colocalization (that is, the posterior probability of the two traits sharing one causal variant at the locus (PP.H4.ABF; 0–1))

multiplied by the sign of colocalization (+1 if both traits have the same risk or higher mean allele and −1 if opposite allele)). **b,** Forest plot representing genetic correlations with SCAD. The Rho coefficient of genetic correlation ($r_g$), obtained using linkage disequilibrium score regression, is represented on the x axis (center of the error bar). The range of each bar represents the 95% CI. Unadjusted P values obtained by two-sided Wald test for genetic correlations are indicated. Asterisks indicate significance after Bonferroni correction for testing 26 traits ($P < 1.9 \times 10^{-3}$) (Supplementary Table 10).

correlation between SCAD and CAD ($r_g = -0.12 \pm 0.04$; $P = 3.7 \times 10^{-3}$) (Supplementary Table 10), including after conditioning SCAD GWAS results on systolic blood pressure (SBP) or diastolic blood pressure (DBP) GWAS results using the multitrait-based conditional and joint analysis (mtCOJO) tool[31] ($r_{gCAD/SBP} = -0.19 \pm 0.04$ ($P = 4.6 \times 10^{-6}$); $r_{gCAD/DBP} = -0.19 \pm 0.04$ ($P = 1.3 \times 10^{-5}$)) (Supplementary Table 12 and Supplementary Fig. 11).

**Cardiovascular risk factors and risk of SCAD and CAD**

We found that SCAD shared several causal variants with SBP and DBP, involving both the same and opposite directional effects (Fig. 3a and Supplementary Table 9). We found one shared locus with hemoglobin levels and a significant genetic correlation with SCAD ($r_g = 0.12 \pm 0.03$; $P = 2.7 \times 10^{-5}$; Fig. 3b). However, SCAD loci were not shared with body mass index (BMI), lipid traits (including low-density lipoprotein (LDL) cholesterol and high-density lipoprotein (HDL)), type 2 diabetes or smoking, and these traits did not correlate with SCAD at the genomic level (Supplementary Tables 9 and 10). Interestingly, we found significant positive genetic correlations both with SBP ($r_g = 0.12 \pm 0.03$; $P = 1.0 \times 10^{-4}$) and DBP ($r_g = 0.17 \pm 0.03$; $P = 2.6 \times 10^{-7}$), indicating a shared genetic basis with SCAD (Fig. 3b and Supplementary Table 10). To assess the extent to which blood pressure and main cardiovascular risk factors may contribute to the risk of SCAD, we leveraged existing GWAS datasets to identify instrumental variables and conducted comparative Mendelian randomization associations with SCAD or CAD. We found robust significant associations estimated by inverse variance-weighted (IVW), MR-Egger and weighted median methods between genetically predicted blood pressure traits and increased risk of SCAD ($\beta_{IVW/SBP} = 0.05 \pm 0.01$ ($P = 7.6 \times 10^{-6}$); $\beta_{IVW/DBP} = 0.10 \pm 0.02$

($P = 1.9 \times 10^{-8}$)) and CAD ($\beta_{IVW/SBP} = 0.04 \pm 0.002$ ($P = 8.6 \times 10^{-49}$); $\beta_{IVW/DBP} = 0.06 \pm 0.004$ ($P = 1.6 \times 10^{-44}$)) (Fig. 4 and Supplementary Table 13). Similar associations were estimated when we analyzed only women with SCAD, women with CAD or men with CAD, although analyses only in men with SCAD were limited by the extremely small numbers of male cases (Supplementary Table 14). Genetically determined BMI, lipid traits, type 2 diabetes and smoking status did not influence the risk for SCAD. However, we were able to confirm that these cardiometabolic traits are strong genetic risk factors for CAD (Fig. 4 and Supplementary Table 13). Our findings indicate that genetically elevated blood pressure is the only shared genetic risk factor between SCAD and CAD, albeit involving potentially different genetic loci.

## Discussion

In this Article, we provide the largest study to date aimed at understanding the genetic basis of SCAD—an understudied cause of AMI that primarily affects women. We report novel associations and demonstrate high polygenic heritability for SCAD. We leverage integrative functional annotations to prioritize genes that are likely to be regulated in VSMCs and the fibroblasts of arteries. Insights from the biological functions of genes highlight the central role of extracellular matrix integrity and reveal impaired tissue coagulation as a novel potential mechanism for SCAD. Globally, we demonstrate the polygenic basis of SCAD to be shared with an important set of cardiovascular diseases. However, a striking directionally opposite genetic impact is found with atherosclerotic CAD, involving multiple risk loci and leading to a genome-wide negative genetic correlation. We provide evidence supporting genetically predicted higher blood pressure as an important

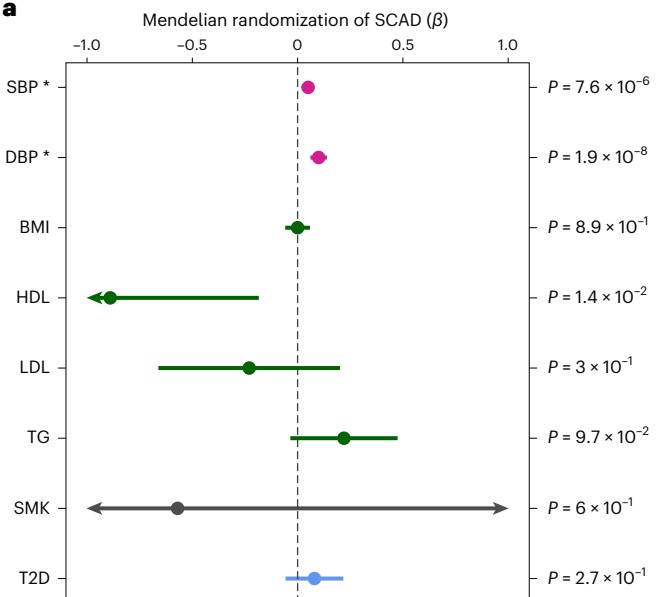

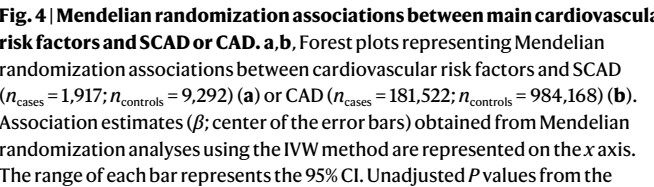

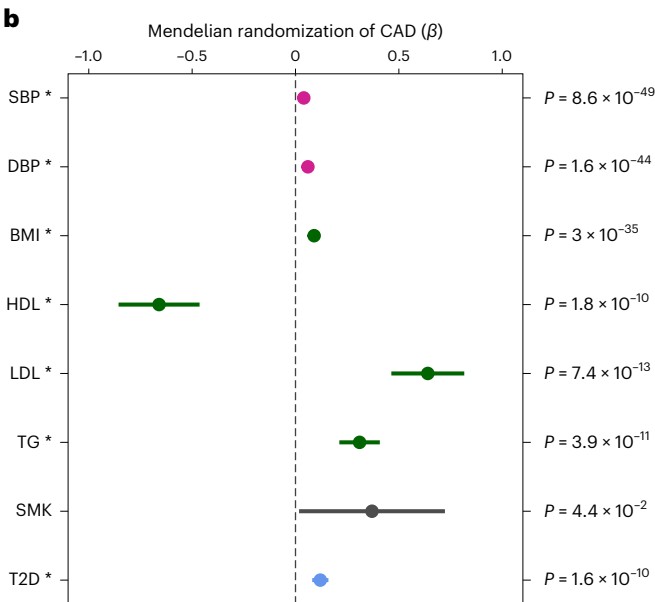

**Fig. 4 | Mendelian randomization associations between main cardiovascular risk factors and SCAD or CAD. a,b,** Forest plots representing Mendelian randomization associations between cardiovascular risk factors and SCAD ($n_{cases}$ = 1,917; $n_{controls}$ = 9,292) (**a**) or CAD ($n_{cases}$ = 181,522; $n_{controls}$ = 984,168) (**b**). Association estimates ($\beta$; center of the error bars) obtained from Mendelian randomization analyses using the IVW method are represented on the x axis. The range of each bar represents the 95% CI. Unadjusted P values from the associations obtained by two-sided Wald test are indicated. $n$ = 340,159 (SBP), 340,162 (DBP), 359,983 (BMI), 315,133 (HDL), 343,621 (LDL), 343,992 (triglycerides (TG)), 164,638 cases and 195,068 controls (smoking (SMK)) and 74,124 cases and 824,006 controls (type 2 diabetes (T2D)). The asterisks indicate significance after Bonferroni correction for testing nine traits ($P < 5.6 \times 10^{-3}$) (Supplementary Table 13).

risk factor for SCAD, but not other well-established cardiovascular factors. Our results set the stage for future investigation of novel biological pathways relevant to both SCAD and CAD and potential therapeutic and preventive strategies specifically targeting SCAD.

As an understudied condition that was previously thought to be uncommon, SCAD was initially suspected to involve rare and highly penetrant mutations. However, recent sequencing studies have suggested that only a small proportion (~3.5%) of SCAD cases are due to rare variants[16,32]. This is in keeping with increasing clinical recognition suggesting that this condition is not rare and occurs globally in populations of both European and non-European ancestry, with similar disease characteristics and probably similar prevalence[2,4,33,34]. Despite a modest sample size, we identified 16 risk loci accounting for about one-quarter of the polygenic heritability, which we estimate to be as high as ~71%, therefore indicating that SCAD is predominantly a complex polygenic disease. However, we acknowledge that larger GWAS settings, including ancestrally diverse populations, will enhance the statistical power needed to provide validation through replication of the reported risk loci and estimated polygenic heritability.

This study supports the presence of genetic overlap between the risk of SCAD and other vascular diseases involving generally younger individuals and more women, such as cervical arterial dissection, migraine, subarachnoid hemorrhage and FMD. These conditions are reported to occur at increased frequency in patients with SCAD[10–13], supporting shared causal biological mechanisms. Among the genes we prioritize as novel SCAD loci, we highlight the ATPase plasma membrane Ca$^{2+}$ transporting 1 gene (*ATP2B1*) that we recently reported to associate with FMD[35]—a well-established locus for blood pressure risk[36] via its role in intracellular calcium homeostasis in VSMCs and blood pressure regulation[37]. Most importantly, we provide evidence for a causal genetic effect of both SBP and DBP in SCAD risk. These findings provide an important genetic basis to support observational data suggesting that control of blood pressure may be an important factor in

reducing the risk of recurrence after SCAD[38]. However, our findings also suggest that controlling other causal risk factors for CAD, such as LDL cholesterol with statins, may confer less benefit in SCAD than in CAD.

Knowledge of the molecular mechanisms leading to SCAD has been limited. Insights from sequencing studies of rare genetic variants have shown that most are associated with genes known from hereditary connective tissue disorders such as vascular Ehlers–Danlos, Loeys–Dietz and Marfan syndromes, as well as adult polycystic kidney disease[16,32]. A striking finding from our study is the identification of the tissue factor gene *F3*—a critical component of tissue-mediated blood coagulation—as a strong candidate gene in a risk locus for SCAD. We found that genetically determined lower expression of *F3* in arterial tissue was associated with a higher risk for SCAD, involving variants located in putative functional regulatory elements in the coronary artery, VSMCs and fibroblasts. Tissue factor is synthesized at the subendothelial level of VSMCs and by fibroblasts in the adventitia surrounding the arteries[39]. In SCAD, once an intramural hemorrhage has initiated, propagation and pressurization of the false lumen may depend, in part, on coagulation and stabilization of the hematoma. Tissue factor is also a druggable target, albeit a potentially challenging one given its known multiple physiological and pathophysiological roles ranging from hemostasis to cancer metastasis. Tissue factor is widely studied in the context of prothrombotic conditions, including atherosclerosis, although notably the genetic variants we describe here do not associate with atherosclerotic disease. This feature is an exception to the highly pleiotropic nature of the variants we describe in the remaining SCAD loci, suggesting impaired tissue-initiated coagulation as a putative specific mechanism in SCAD.

We identify regulation of the extracellular matrix of arteries as the predominant polygenic biological mechanism for SCAD. Integrative prioritization analyses revealed 13 potential causal genes with established key roles in maintaining arterial wall integrity and function. Among these, we highlight the serine protease HTRA1 and

metallopeptidase inhibitor TIMP3, which are involved in matrix disassembly. *TIMP3* clusters in the main network for extracellular matrix organization that includes *ADAMTSL4*, *LRP1* and *COL4A1*, with connections with subnetworks of *F3*. This clustering is consistent with the biological function of TIMP3 as an inhibitor of matrix metalloproteinases with domains interacting with ADAMTS proteins and LRP1, involving proteins encoded by genes prioritized in SCAD loci[40]. Interestingly, we found a novel association signal with SCAD in the metallopeptidase thrombospondin type 1 domain containing 4 gene (*THSD4*) that promotes fibrillin 1 elastic fiber assembly, and confirm the previously reported associations near *ADAMTSL4* and *FBN1* (refs. [18],[20]). We showed that genetically decreased expressions of these genes in arteries were correlated with higher SCAD risk alleles in arteries or fibroblasts. This finding suggests that a genetic predisposition to a weaker extracellular matrix may increase the vulnerability of traversing intramural microvessels to disruption, increasing the risk of initiation and propagation of a false lumen within the coronary vessel wall, leading to SCAD.

Many of the risk loci for SCAD that we report here, as well as their prioritized genes, are already known from atherosclerotic disease GWASs. However, here we provide compelling and intriguing evidence for the opposite directionality of a substantial fraction of genetic bases for SCAD versus CAD, suggesting that some key biological mechanisms involved in the two diseases are also likely to be opposite, which is consistent with the clinical observation of a lower-than-expected burden of atherosclerotic disease in patients with SCAD. For example, the association signals in the *COL4A1/COL4A2* locus are in an opposite direction to their contribution to CAD[41]. This locus encodes α1 and α2 chains of type IV collagen, with transcripts generated through a common promoter. Type IV collagen is the main component of the basement membrane of arterial cells and plays a key role in the structural integrity and biological functions of VSMCs in the tunica muscularis. Decreased collagen IV expression increases the risk of CAD[15,42]. Proposed potential mechanisms for this include a disinhibition of VSMC-intimal migration during atherogenesis or an increase in the vulnerability of atherosclerotic plaque to rupture[42]. In contrast with CAD, our data indicate that genetically mediated increased collagen IV expression also increases the risk of SCAD. Better understanding of how these directionally opposite changes modify the risk of CAD and SCAD has considerable potential to enhance our understanding of the molecular genetic mechanisms that confer risk in both diseases.

## Online content

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

David Adlam [1,2,41], Takiy-Eddine Berrandou [3,4,41], Adrien Georges [3,41], Christopher P. Nelson [1,2], Eleni Giannoulatou[5,6], Joséphine Henry[3], Lijiang Ma[7], Montgomery Blencowe[8,9], Tamiel N. Turley[10], Min-Lee Yang[11,12,13], Sandesh Chopade[14,15], Chris Finan [14,15], Peter S. Braund [1,2], Ines Sadeg-Sayoud[3], Siiri E. Iismaa [5,6], Matthew L. Kosel[16], Xiang Zhou [17], Stephen E. Hamby [1,2], Jenny Cheng[8,9], Lu Liu[3], Ingrid Tarr[5], David W. M. Muller [5,6,18], Valentina d'Escamard[19], Annette King[20], Liam R. Brunham [21], Ania A. Baranowska-Clarke[1,2], Stéphanie Debette [22], Philippe Amouyel [23], Jeffrey W. Olin[20], Snehal Patil[17], Stephanie E. Hesselson [5,6], Keerat Junday[5,6], Stavroula Kanoni [24], Krishna G. Aragam [25,26,27,28], Adam S. Butterworth [29,30,31], CARDIoGRAMPlusC4D*, MEGASTROKE*, International Stroke Genetics Consortium (ISGC) Intracranial Aneurysm Working Group*, Marysia S. Tweet[32], Rajiv Gulati[32], Nicolas Combaret [33], DISCO register*, Daniella Kadian-Dodov[20], Jonathan M. Kalman[34,35], Diane Fatkin [5,6,18], Aroon D. Hingorani [14,15], Jacqueline Saw [36], Tom R. Webb [1,2], Sharonne N. Hayes [32], Xia Yang [8,9,37,38], Santhi K. Ganesh[11,13], Timothy M. Olson [32,39], Jason C. Kovacic [5,6,18,19,20], Robert M. Graham [5,6,18], Nilesh J. Samani[1,2] & Nabila Bouatia-Naji [3]

[1]Department of Cardiovascular Sciences, Glenfield Hospital, Leicester, UK. [2]NIHR Leicester Biomedical Research Centre, Glenfield Hospital, Leicester, UK. [3]Université Paris Cité, Paris Cardiovascular Research Center, Inserm, Paris, France. [4]Quantitative Genetics and Genomics, Aarhus University, Aarhus, Denmark. [5]Victor Chang Cardiac Research Institute, Sydney, New South Wales, Australia. [6]School of Clinical Medicine, Medicine and Health, University of New South Wales, Sydney, New South Wales, Australia. [7]Department of Genetics and Genomic Sciences, Icahn School of Medicine at Mount Sinai, New York, NY, USA. [8]Department of Integrative Biology and Physiology, University of California, Los Angeles, Los Angeles, CA, USA. [9]Interdepartmental Program of Molecular, Cellular, and Integrative Physiology, University of California, Los Angeles, Los Angeles, CA, USA. [10]Mayo Clinic Graduate School of Biomedical Sciences, Mayo Clinic, Rochester, MN, USA. [11]Division of Cardiovascular Medicine, Department of Internal Medicine, University of Michigan Medical School, Ann Arbor, MI, USA. [12]Department of Computational Medicine and Bioinformatics, University of Michigan, Ann Arbor, MI, USA. [13]Department of Human Genetics, University of Michigan Medical School, Ann Arbor, MI, USA. [14]Institute for Cardiovascular Science, University

College London, London, UK. [15]British Heart Foundation Research Accelerator, University College London, London, UK. [16]Department of Quantitative Health Sciences, Mayo Clinic, Rochester, MN, USA. [17]Department of Biostatistics, University of Michigan School of Public Health, Ann Arbor, MI, USA. [18]Cardiology Department, St Vincent's Hospital, Sydney, New South Wales, Australia. [19]Cardiovascular Research Institute, Icahn School of Medicine at Mount Sinai, New York, NY, USA. [20]Zena and Michael A. Wiener Cardiovascular Institute and Marie-Josée and Henry R. Kravis Center for Cardiovascular Health, Icahn School of Medicine at Mount Sinai, New York, NY, USA. [21]Centre for Heart Lung Innovation, Departments of Medicine and Medical Genetics, University of British Columbia, Vancouver, British Columbia, Canada. [22]Department of Neurology, Bordeaux University Hospital, Inserm, Bordeaux, France. [23]Université de Lille, Inserm, CHU Lille, Institut Pasteur de Lille, RID-AGE - Labex DISTALZ - Risk Factors and Molecular Determinants of Aging-Related Disease, Lille, France. [24]William Harvey Research Institute, Barts and the London School of Medicine and Dentistry, Queen Mary University of London, London, UK. [25]Cardiovascular Research Center, Massachusetts General Hospital, Boston, MA, USA. [26]Center for Genomic Medicine, Massachusetts General Hospital, Boston, MA, USA. [27]Cardiovascular Disease Initiative, Broad Institute of MIT and Harvard, Cambridge, MA, USA. [28]Program in Medical and Population Genetics, Broad Institute of MIT and Harvard, Cambridge, MA, USA. [29]British Heart Foundation Cardiovascular Epidemiology Unit, Department of Public Health and Primary Care, University of Cambridge, Cambridge, UK. [30]Health Data Research UK Cambridge, Wellcome Genome Campus and University of Cambridge, Cambridge, UK. [31]British Heart Foundation Centre of Research Excellence, Division of Cardiovascular Medicine, Addenbrooke's Hospital, Cambridge, UK. [32]Department of Cardiovascular Medicine, Mayo Clinic, Rochester, MN, USA. [33]Department of Cardiology, CHU Clermont-Ferrand, CNRS, Université Clermont Auvergne, Clermont-Ferrand, France. [34]Department of Cardiology, Royal Melbourne Hospital, Melbourne, Victoria, Australia. [35]Department of Medicine, University of Melbourne, Melbourne, Victoria, Australia. [36]Vancouver General Hospital, Division of Cardiology, University of British Columbia, Vancouver, British Columbia, Canada. [37]Institute for Quantitative and Computational Biosciences, University of California, Los Angeles, Los Angeles, CA, USA. [38]Molecular Biology Institute, University of California, Los Angeles, Los Angeles, CA, USA. [39]Department of Pediatric and Adolescent Medicine, Mayo Clinic, Rochester, MN, USA. [41]These authors contributed equally: David Adlam, Takiy-Eddine Berrandou, Adrien George. *Lists of authors and their affiliations appear at the end of the paper. ✉e-mail: da134@leicester.ac.uk; nabila.bouatia-naji@inserm.fr

## CARDIoGRAMPlusC4D

Stavroula Kanoni[24], Krishna G. Aragam[25,26,27,28], Adam S. Butterworth[29,30,31] & Nilesh J. Samani[1,2]

Full lists of members and their affiliations appear in the Supplementary Information.

## MEGASTROKE

Stéphanie Debette[22] & Philippe Amouyel[23]

Full lists of members and their affiliations appear in the Supplementary Information.

## International Stroke Genetics Consortium (ISGC) Intracranial Aneurysm Working Group

Mark K. Bakker[40] & Ynte M. Ruigrok[40]

[40]Department of Neurology and Neurosurgery, University Medical Center Utrecht Brain Center, Utrecht University, Utrecht, the Netherlands.

## DISCO register

Nicolas Combaret[33]

Full lists of members and their affiliations appear in the Supplementary Information.

## Methods

### Patients and control populations

Our meta-analysis included participants of European ancestry from eight studies: DISCO-3C, SCAD-UK I, SCAD-UK II, Mayo Clinic, DEFINE-SCAD, CanSCAD/MGI, VCCRI I and VCCRI II (Supplementary Fig. 1). Patients with SCAD presented with similar clinical characteristics (Supplementary Table 1), as well as homogeneous diagnosis, exclusion and inclusion criteria. All of the studies were approved by national and/or institutional ethical review boards. Further study-specific clinical details are provided in the Supplementary Note.

### Genome-wide association meta-analysis

Details of the pre-imputation quality control steps for each study are listed in Supplementary Table 15. Briefly, genotyping was performed using commercially available arrays or genome sequencing (SCAD-UK II and VCCRI II). To increase the number of tested SNPs and the overlap of variants available for analysis between different arrays, the genotypes of all European ancestry cohorts except SCAD-UK II and VCCRI II were imputed to the Haplotype Reference Consortium version 1.1 reference panel[45] on the Michigan Imputation Server[46]. A GWAS was conducted in each study under an additive genetic model using PLINK version 2.0 (ref. [47]). For chromosome X, males and females were both on a 0.2 scale under the chromosome X inactivation assumption model. Models were adjusted for population structure using residues from the first five principal components and sex, except in the women-only analyses. Before meta-analysis, we removed SNPs with low minor allele frequencies (<0.01), low imputation quality ($r^2 < 0.8$) and deviations from Hardy–Weinberg equilibrium ($P < 10^{-5}$). A total of 6,691,677 variants met these criteria and were kept in the final results.

Results from individual GWASs were combined using an inverse variance-weighted fixed-effects meta-analysis in METAL software[48], with correction for genomic control. Heterogeneity was assessed using the $I^2$ metric from the complete study-level meta-analysis. Between-study heterogeneity was tested using Cochran's $Q$ statistic and considered significant at $P \leq 10^{-3}$. The genome-wide significance threshold was set at the level of $P = 5.0 \times 10^{-8}$. LocusZoom (http://locuszoom.org/) was used to provide regional visualization of the results.

### Functional annotation

**Identification of potential functional variants.** To generate a list of potential functional variants, we first identified the 95% credible set of variants using the ppfunc function of the corrcoverage R package (version 1.2.1). The posterior probability of causality was evaluated from marginal $z$ scores for all variants within 500 kilobases (kb) of the lead SNP at each locus. In the *COL4A1/COL4A2* locus, where we found two association signals, these were separated by placing an equidistant border from each lead SNP for the inclusion of SNPs in the analysis. Variants with a cumulated posterior probability of up to 95% were kept for further analyses. To consider potentially poorly imputed variants in one of the individual case–control studies, we also included variants in high linkage disequilibrium ($r^2 > 0.7$) with the lead SNP at each locus, based on information from European populations (1000 Genomes reference panel) queried using the ldproxy function of the LDlinkR package (version 1.1.2)[49].

**Enrichment of SCAD variants in regulatory regions.** To calculate the enrichment of SCAD-associated SNPs among functionally annotated genomic regions, we retrieved available H3K27ac chromatin immunoprecipitation followed by sequencing (ChIP-seq) datasets (narrowPeak beds) in any tissue from ENCODE (https://www.encodeproject.org/ (ref. [50])) and single-nucleus assay for transposase-accessible chromatin with sequencing (snATAC-seq) peak files (bed format) from the Human Enhancer Atlas (http://catlas.org/humanenhancer (ref. [24])). A complete list of datasets is available in Supplementary Table 16. For H3K27ac marks, bed files corresponding to the same tissue were concatenated and sorted before combining overlapping peaks using the bedtools (version 2.29.0) merge command. Variant enrichment was calculated using the GREGOR package (version 1.4.0)[43]. All potential functional variants (95% credible set and linkage disequilibrium proxies as described above) were used as inputs and the parameters were adjusted so as not to pick additional linkage disequilibrium proxies (LDWINDOWSIZE = 1). $P$ values were adjusted for multiple testing by the application of Bonferroni correction.

**Identification of variants with potential regulatory function.** We used H3K27ac peaks in coronary arteries (as described above), open chromatin regions in healthy coronary arteries (obtained as previously described[35,51]) and open chromatin regions from merged snATAC-seq clusters, which were mapped fragments from snATAC-seq in 25 adult tissues that we retrieved from the Gene Expression Omnibus (GSE184462)[24] in bed format. Mapped fragments from all clusters representing >1% of cells in at least one arterial tissue (T lymphocyte 1, CD8$^+$, endothelial general 2, endothelial general 1, macrophage general, fibroblast general, vascular smooth muscle 2 or vascular smooth muscle 1) were extracted and grouped by annotated cell type as T lymphocytes, macrophages, fibroblasts, endothelial cells and VSMCs, respectively. Genome coverage was calculated using the bedtools (version 2.29.0) coverage function. We detected peaks from bedGraph output using the MACS2 bdgpeakcall function (Galaxy Version 2.1.1.20160309.0) on the Galaxy webserver[52,53]. All peak files were extended 100 base pairs upstream and downstream using the bedtools (version 2.29.0) slop function. We detected overlaps of SCAD potential functional variants with relevant genomic regions using the findOverlap function from the rtracklayer package (version 1.52.1)[54]. We used the Integrated Genome Browser (version 9.1.8) to visualize read density profiles and peak positions in the context of the human genome[55].

**Gene prioritization.** Genes located within 500 kb of lead variants were annotated to prioritize the most likely causal genes. To find the closest gene(s) from lead SNPs and genes overlapping with variants in the credible set of causal SNPs, gene coordinates were retrieved from Gencode release 38 and aligned to hg19 genomic coordinates (gencode.v38lift37.annotation.gff3.gz). Significant eQTL associations and all SNP–gene eQTL associations in the version 8 release of the GTEx database were retrieved from the GTEx website (www.gtexportal.org/home/datasets). Colocalization of association with SCAD and eQTLs was evaluated using the R coloc package (version 5.1.0) with default values as priors. We considered that there was evidence for colocalization if H4 coefficients were >75% or if eQTL association was significant for SCAD lead SNPs and H4 was over 25%. TWASs were performed using the FUSION R/Python package[44]. Gene expression models were pre-computed from GTEx data (version 8 release) and were provided by the authors. Only genes with a heritability $P < 0.01$ were used in the analysis. Both tools used linkage disequilibrium information from the European panel of phase 3 of the 1000 Genomes Project. Bonferroni multiple testing correction was applied using the p.adjust function in R (version 4.1.0). Significant capture Hi-C hits in aorta tissue were provided as supplementary data by Jung et al.[25]. Genes associated with mouse cardiovascular phenotypes (code MP:0005385) were retrieved from the Mouse Genome Informatics database (www.informatics.jax.org)[56]. We also queried the DisGeNET database, using the disgenet2r package (version 0.99.2), for genes with reported evidence in human cardiovascular disease (code C14) with a score of >0.2, including "ALL" databases[57]. In the absence of a missense variant, colocalization and TWAS criteria were given a tenfold weight compared with other criteria. At each locus, we prioritized genes fulfilling the largest number of criteria. In cases where several candidates were retained, we prioritized genes that were most likely to have a function in arterial disease (for example, expression in arterial tissues or exclusion of pseudo-genes).

**Druggability of prioritized genes.** The druggability of the gene products identified through the GWAS was assessed by reference to the set of genes encoding druggable targets derived by Finan et al.[26] using ChEMBL version 17. Targets in this set are subclassified into: (1) the efficacy targets of approved agents and clinical phase drug candidates (tier 1); (2) genes encoding targets with known bioactive drug-like small molecule binding partners and those with substantial sequence with approved drug targets (tier 2); and (3) genes encoding secreted or extracellular proteins, proteins with more distant similarity to approved drug targets and members of key druggable gene families not already included in tiers 1 or 2. Further lookups of approved and clinical phase targets were performed against ChEMBL[58] version 30 and the British National Formulary (accessed 9 April 2021). Note that identified drug targets can either be: (1) a single protein providing a 1:1 link with the causal gene nominated in a GWAS and post-GWAS analysis; (2) a protein complex where the causal gene can encode a member of the complex; or (3) a protein family with the causal gene being a member of the family.

**Bayesian network query of SCAD candidate genes.** Gene expression data from the aorta artery, coronary artery, tibial artery and cultured fibroblasts were curated from version 8 of the GTEx database (ref. 28). Gene expression data from the mouse aorta was curated from the Hybrid Mouse Diversity Panel (HMDP)[27]. Tissue-specific gene regulatory Bayesian networks were constructed from the GTEx and HMDP gene expression data using RIMBANET[29]. The Bayesian network from each dataset included only network edges that passed a probability of >30% across 1,000 generated Bayesian networks starting from different random genes. Bayesian networks were combined for the top GWAS hits query, and mouse gene symbols were converted to their human orthologs. Bayesian networks were queried for the identified top GWAS hits to identify their first-degree network connections and to determine connections between their surrounding subnetwork nodes. The directions of edges were informed by prior knowledge, such as eQTLs and previously known regulatory relationships between genes. Subnetworks were annotated by top biological pathways representative of the subnetwork genes using Enrichr with a false discovery rate of <0.05.

### Colocalization with other traits and diseases
Summary statistics were retrieved from individual studies, as indicated in Supplementary Table 17. At each locus, we selected variants found in both SCAD and the other studies with a high quality of imputation ($r^2 > 0.9$) and located within 500 kb from the SCAD lead SNP. *COL4A1* and *COL4A2* loci were separated by placing an equidistant border from SCAD lead SNPs for the inclusion of SNPs in the analysis. Signal colocalization was evaluated using the R coloc package (version 5.1.0) with default values as priors. We reported H4 coefficients indicating the probability of two signals sharing a common causal variant at each locus.

### Heritability estimates and genetic correlation
We used linkage disequilibrium score regression[21] implemented in the ldsc package (version 1.0.1; https://github.com/bulik/ldsc/) and SumHer[22] implemented in the LDAK software (www.ldak.org) to quantify the heritability explained by common variants or SNP-based heritability ($h^2_{SNP}$) for SCAD and the degree of genetic correlation between SCAD and other diseases and traits. We also used SumHer to estimate the SNP-based heritability attributable to loci associated with SCAD at genome-wide statistical significance. Loci were defined as the 1 megabase region around lead SNPs in the GWAS meta-analysis. SNPs belonging to each locus were used as annotations to calculate the partitioned heritability. Two analyses were performed: one that considered separated loci and a second that aggregated all SNPs as one annotation. Summary statistics were acquired from the respective consortia and are detailed in Supplementary Table 17. For each trait,

we refined the summary statistics to the subset of HapMap 3 SNPs to reduce the potential bias due to poor imputation quality. Correlation analyses were restricted to European ancestry meta-analyses summary statistics. We used the European linkage disequilibrium score files calculated from the 1000 Genomes reference panel and provided by the developers. $P < 1.9 \times 10^{-3}$, corresponding to adjustment for 26 independent phenotypes, was considered significant. We conditioned SCAD association on cardiometabolic trait genetic association using the mtCOJO tool from the GCTA pipeline[31]. The resulting summary statistics were then used to compute genetic correlations between SCAD, conditioned on cardiometabolic traits and traits of interest.

### Mendelian randomization analyses
We applied a stringent selection process for instrumental variables to ensure the validity of our Mendelian randomization results. To select valid instrumental variables that respect the three key assumptions ((1) strong association with the exposure; (2) independence from potential confounders between the exposure and outcome; and (3) influence on the outcome only through the exposure), we used linkage disequilibrium clumping with a $P$ value threshold of $<5 \times 10^{-8}$ and a linkage disequilibrium $r^2 < 0.001$ within a 10,000 kb window based on the European population in the 1000 Genomes Project. We excluded candidate instrumental variables that were absent in the summary statistics data from a GWAS of our outcome (SCAD/CAD). To minimize the risk of horizontal pleiotropy, we removed candidate instrumental variables that were associated with the outcome or in high to moderate linkage disequilibrium ($r^2 > 0.6$ within a 10,000 kb window).

We used the multiplicative random-effects IVW method[59] implemented in the TwoSampleMR R package to estimate the associations of genetically predicted cardiovascular risk factors, including blood pressure (SBP and DBP), lipids (HDL, LDL and triglycerides), BMI, smoking liability and type 2 diabetes, with each of the outcomes of interest (SCAD or CAD). Estimates were scaled to a doubling in genetically predicted smoking risk, or to a one-unit increase in the genetically predicted trait for the continuous traits. We performed sensitivity analyses using the weighted median and MR-Egger methods to assess the consistency of estimates under alternative assumptions about genetic pleiotropy, as recommended[59]. We also performed Cochran's $Q$ test to assess the heterogeneity between estimates obtained using different variants. As 11 risk factors were assessed, a Bonferroni-corrected significance level of $0.05/9 = 5.6 \times 10^{-3}$ was used as the threshold for statistical significance in this analysis. $P$ values between $5.6 \times 10^{-3}$ and 0.05 were considered suggestively significant.

### Reporting summary
Further information on research design is available in the Nature Portfolio Reporting Summary linked to this article.

## Data availability
Gene reference names and coordinates were retrieved from the GENCODE project through the European Bioinformatics Institute FTP server. gencode.v38.annotation.gff3 and gencode.v38lift37.annotation.gff3 files were used. eQTL data were retrieved from version 8 of the GTEx database (https://gtexportal.org/home/datasets). H3K27ac ChIP-seq datasets (narrowPeak beds) in any tissue were retrieved from ENCODE (https://www.encodeproject.org/). Single-nucleus ATAC-seq peak files (bed format) were retrieved from the Human Enhancer Atlas (http://catlas.org/humanenhancer). Open chromatin regions in healthy coronary arteries were generated from raw reads retrieved from the Sequence Read Archive (SRR2378591, SRR2378592 and SRR2378593). Raw snATAC-seq data in 25 adult tissues were retrieved from the Gene Expression Omnibus (GSE184462). Gene expression models for TWASs were retrieved from the Gusev laboratory website (http://gusevlab.org/projects/fusion/) based on GTEx data (v8 release). Gene expression data from aorta arteries, coronary arteries, tibial arteries and cultured

fibroblasts were curated from version 8 of the GTEx database (www.gtexportal.org/home/datasets). Gene expression data from mouse aortas were curated from the HMDP. Genes associated with mouse cardiovascular phenotypes (code MP:0005385) were retrieved from Mouse Genome Informatics (www.informatics.jax.org). GWAS summary statistics were retrieved from http://www.cardiogramplusc4d.org/data-downloads/, http://ftp.ebi.ac.uk/pub/databases/gwas/summary_statistics/, https://www.megastroke.org/, http://www.nealelab.is/uk-biobank or https://diagram-consortium.org/downloads.html or retrieved from authors, as detailed in Supplementary Table 17. The set of genes encoding druggable targets was derived using ChEMBL version 17 and further analyzed using ChEMBL version 30 and the British National Formulary (accessed 9 April 2021). Summary statistics for SCAD association from the meta-analysis are available in the GWAS Catalog (GCP000522). Full lists of the datasets used in this study, along with the corresponding accession numbers, are available in Supplementary Tables 16 and 17.

## Code availability

Publicly available software and packages were used throughout this study according to each developer's instructions. No custom algorithms were generated for the study. The following software was used and can be downloaded or accessed online: Michigan Imputation Server (https://imputationserver.sph.umich.edu/index.html#), PLINK version 2.0 (https://www.cog-genomics.org/plink/2.0/), METAL (March 2011 version; http://csg.sph.umich.edu/abecasis/Metal/), LocusZoom (version 0.12.0; http://locuszoom.org/), FUSION (version released 15 November 2021; https://github.com/gusevlab/fusion_twas), R (version 4.1.0; https://cran.r-project.org/), RStudio (version 1.2.335; https://posit.co/download/rstudio-desktop/), R packages (colorspace_2.0–3, ggnewscale_0.4.7, corrcoverage_1.2.1, locuscomparer_1.0.0, coloc_5.1.0, dplyr_1.0.8, tidyr_1.2.0, ggrepel_0.9.1, RColorBrewer_1.1-3, shades_1.4.0, rtracklayer_1.52.1, LDlinkR_1.1.2, ggplot2_3.3.5, GenomicRanges_1.44.0, GenomeInfoDb_1.28.4, IRanges_2.26.0, S4Vectors_0.30.2, BiocGenerics_0.38.0 and readr_2.1.2; available from https://cran.r-project.org/ and/or https://www.bioconductor.org/), disgenet2r_0.99.2 (https://www.disgenet.org/disgenet2r), bedtools (version 2.29.0; https://github.com/arq5x/bedtools2), Galaxy (https://usegalaxy.org/), the Integrated Genome Browser (version 9.1.8; https://www.bioviz.org/), GREGOR (version 1.4.0; http://csg.sph.umich.edu/GREGOR/), the ldsc package (version 1.0.1; https://github.com/bulik/ldsc/), SumHer implemented in the LDAK software (version 5.2; www.ldak.org), the mtCOJO tool from the GCTA pipeline (version 1.94.1; https://yanglab.westlake.edu.cn/software/gcta/), TwoSampleMR (version 0.5.6; https://mrcieu.github.io/TwoSampleMR/), RIMBANET (version from 26 June 2019; https://labs.icahn.mssm.edu/zhulab/?s=rimbanet) and the Enrichr webserver (version from 29 March 2021; https://maayanlab.cloud/Enrichr/). The code for the druggability analysis is available from GitLab (https://cfinan.gitlab.io/biomisc/scripts/drug_lookups.html). Scripts used for the analyses were deposited in our repository (https://github.com/takiy-berrandou/GWAS-meta-analysis-of-SCAD-paper-analysis-scripts-). Specific options are indicated in the Methods where relevant.

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

## Acknowledgements

We acknowledge clinical colleagues who referred SCAD cases, patients who supported this research and healthy volunteers included in this study. This study was funded by a European Research Council grant (ERC-Stg-ROSALIND-716628 to N.B.-N.); the French Society of Cardiology, through Fondation Coeur et Recherche (to N.B.-N.); La Fédération Française de Cardiologie (to N.B.-N.); the British Heart Foundation (PG/13/96/30608 to D.A. and SP/16/4/32697 to T.R.W.); the Leicester NIHR Biomedical Research Centre and BeatSCAD charity (to D.A.); the National Health and Medical Research Council (to R.M.G.); the Cardiac Society of Australia and New Zealand Cardiovascular Research Innovation Grant Australia (APP1161200 to R.M.G., D.W.M.M., D.F. and J.C.K.); the New South Wales Health Early-Mid Career Cardiovascular Grant (to E.G.); the New South Wales Health Senior Scientist Cardiovascular Grant (RG194194 to J.C.K.); the New South Wales Health Senior Clinician Cardiovascular Grant (RG193092 to R.M.G.); Bourne Foundation, Agilent and SCAD Research funds (to M.S.T., R.G., S.N.H. and T.M.O.); the National Institutes of Health (T32 GM72474 to T.M.O., R35HL161016 to S.K.G. (who was also supported by R01HL086694 and R01HL139672), R01HL148167 to J.C.K. (to support J.W.O., D.K.-D. and V.d.E.), R01HL147883 to X.Y. and 1K23HL155506 to M.S.T.); the Genome Consortia and Mayo Clinic Center for Individualized Medicine of the Heart and Stroke Foundation of Canada (G-17-0016340 to J.S.); the Canadian Institutes of Health Research (grant 136799 to J.S.); the US Department of Defense; the University of Michigan Frankel Cardiovascular Center M-BRISC program (to S.K.G.); the University of Michigan A. Alfred Taubman Institute (to S.K.G.); a Michael Smith Foundation for Health Research Scholar award (to J.S. and L.R.B.); the FMD Society of America; the American Heart Association (pre-doctoral fellowship 829009 to M.B.); and the UCLA Integrative Biology and Physiology Edith Hyde fellowship (to M.B.). Genotyping of DISCO and SCAD-UK II patients was performed in a platform supported by the Spanish National Cancer Research Centre, at the Human Genotyping laboratory, a member of the CeGen Biomolecular resources platform (PRB3),

supported by grant PT17/0019, of the PE I+D+i 2013–2016, funded by Instituto de Salud Carlos III and the European Regional Development Fund. Fondation Alzheimer supported genotyping of the 3C study (Paris, France) to P.A. We thank AstraZeneca's Centre for Genomics Research (Discovery Sciences, BioPharmaceuticals R&D) for funding the sequencing of participants in cohort SCAD-UK I and providing bioinformatics support. We acknowledge the leadership of the ESC-ACVC SCAD Study Group. The DISCO investigators thank the French Society of Cardiology and French Coronary Atheroma and Interventional Cardiology Group for support, as well as clinical research associates of the Clermont-Ferrand University Hospital: E. Chazot, C. Bellanger, L. Cubizolles, A. Thalamy and O. Lamallem. The SCAD-UK study investigators acknowledge J. Middleton, J. Plume, D. Alexander, D. Lawday and A. Marshall for support with SCAD research, as well as the Research Analytics and Informatics team at AstraZeneca's Centre for Genomics for processing and analyzing sequencing data. The VCCRI study investigators thank C. M. Y. Wong, K. Mishra and R. Johnson for contributions to data collection and sample processing, as well as the Medical Genome Reference Bank, including the 45 and Up and ASPREE study patients who were controls for this study. The CanSCAD/MGI study investigators acknowledge the University of Michigan Precision Health Initiative and Medical School Central Biorepository for providing biospecimen storage, management, processing and distribution services, as well as the Center for Statistical Genetics in the Department of Biostatistics at the School of Public Health for genotype data management in support of this research. The MEGASTROKE project received funding from sources specified at http://www.megastroke.org/acknowledgements.html. The GTEx Project was supported by the Common Fund of the Office of the Director of the National Institutes of Health, as well as by the National Cancer Institute, National Human Genome Research Institute, National Heart, Lung, and Blood Institute, National Institute on Drug Abuse, National Institute of Mental Health and National Institute of Neurological Disorders and Stroke. We acknowledge the FMD Society of America and Vancouver SCAD Conference organizers for enabling study enrollments at patient meetings.

## Author contributions

D.A., T.-E.B., A.G., N.J.S. and N.B.-N. wrote the manuscript. D.A., T.-E.B., A.G., C.P.N., E.G., M.S.T., R.G., A.D.H., D.K.-D., J.S., S.N.H., X.Y., S.K.G., T.M.O., J.C.K., R.M.G., N.J.S. and N.B.-N. designed the study and conceived of the analyses. D.A., A.G., C.P.N., L.M., T.N.T., M.-L.Y., P.S.B., S.E.I., M.L.K., S. E. Hamby, L.L., V.d.E., A.A.B.-C., K.J., T.R.W., P.A., S.K.G., T.M.O., J.M.K. and N.B.-N. performed the genotyping. D.A., I.T., D.W.M.M., V.d.E., A.K., L.R.B., S.D., P.A., J.W.O., S. E. Hesselson, M.S.T., R.G., N.C., D.K.-D., J.S., L.R.B., S.K.G., J.M.K., D.F., S.N.H., J.C.K. and R.M.G. contributed samples/phenotypes. T.-E.B., A.G., C.P.N., E.G., J.H., L.M., M.B., M.-L.Y., S.C., C.F., I.S.-S., M.L.K., X.Z., J.C., I.T., S.P., S.K., K.G.A., A.S.B., X.Y., J.C.K. and N.B.-N. analyzed the data. D.A., T.-E.B., A.G., J.H., T.N.T., S.K.G., M.S.T., R.M.G., T.R.W., S.N.H., X.Y., T.M.O., J.C.K., N.J.S. and N.B.-N. edited the manuscript.

## Competing interests

D.A. has received kind support from AstraZeneca (for gene sequencing in patients with SCAD) and grant funding from AstraZeneca for unrelated research. He has also received research funding from Abbott Vascular to support a clinical research fellow and has undertaken consultancy for General Electric to support research funds. He holds unrelated patents EP3277337A1 and PCT/GB2017/050877. The remaining authors declare no competing interests.

## Additional information

**Correspondence and requests for materials** should be addressed to David Adlam or Nabila Bouatia-Naji.

# Reporting Summary

## Statistics

For all statistical analyses, confirm that the following items are present in the figure legend, table legend, main text, or Methods section.

| n/a | Confirmed | |
|---|---|---|
| ☐ | ☒ | The exact sample size (*n*) for each experimental group/condition, given as a discrete number and unit of measurement |
| ☒ | ☐ | A statement on whether measurements were taken from distinct samples or whether the same sample was measured repeatedly |
| ☐ | ☒ | The statistical test(s) used AND whether they are one- or two-sided *Only common tests should be described solely by name; describe more complex techniques in the Methods section.* |
| ☐ | ☒ | A description of all covariates tested |
| ☐ | ☒ | A description of any assumptions or corrections, such as tests of normality and adjustment for multiple comparisons |
| ☐ | ☒ | A full description of the statistical parameters including central tendency (e.g. means) or other basic estimates (e.g. regression coefficient) AND variation (e.g. standard deviation) or associated estimates of uncertainty (e.g. confidence intervals) |
| ☐ | ☒ | For null hypothesis testing, the test statistic (e.g. *F*, *t*, *r*) with confidence intervals, effect sizes, degrees of freedom and *P* value noted *Give P values as exact values whenever suitable.* |
| ☐ | ☒ | For Bayesian analysis, information on the choice of priors and Markov chain Monte Carlo settings |
| ☒ | ☐ | For hierarchical and complex designs, identification of the appropriate level for tests and full reporting of outcomes |
| ☒ | ☐ | Estimates of effect sizes (e.g. Cohen's *d*, Pearson's *r*), indicating how they were calculated |

*Our web collection on statistics for biologists contains articles on many of the points above.*

## Software and code

Policy information about availability of computer code

| Data collection | No software was used for data collection. |
|---|---|
| Data analysis | 1. For genotyped cohorts, genotypes were imputed to HRC v1.1 reference panel on the Michigan Imputation Server. |
| | 2. GWAS was conducted in each study under an additive genetic model using PLINK v2.0. |
| | 3. Meta-analysis was performed using METAL software (March 2011 version). |
| | 4. LocusZoom 0.12.0 (http://locuszoom.org/) was used to provide regional visualization of results on April 14th 2022. |
| | 5. TWAS was performed using FUSION R/python package (version released 15th November 2021). |
| | 6. Statistical analyses and plotting were performed using R (4.1.0), through RStudio interface software (v1.2.335). Following packages were used: colorspace_2.0-3, ggnewscale_0.4.7, corrcoverage_1.2.1, disgenet2r_0.99.2, locuscomparer_1.0.0, coloc_5.1.0, dplyr_1.0.8, tidyr_1.2.0, ggrepel_0.9.1, RColorBrewer_1.1-3, shades_1.4.0, rtracklayer_1.52.1, LDlinkR_1.1.2, ggplot2_3.3.5, GenomicRanges_1.44.0, GenomeInfoDb_1.28.4, IRanges_2.26.0, S4Vectors_0.30.2, BiocGenerics_0.38.0, readr_2.1.2. |
| | 7. bedtools (v2.29.0) was used to handle bed files, merge peak files from the same tissue, generate bedGraph files for visualization of snATAC-Seq datasets. |
| | 8. snATAC-Seq peaks were detected using MACS2 bdgpeakcall function (Galaxy Version 2.1.1.20160309.0) on Galaxy webserver (https://usegalaxy.org/). |
| | 9. We used Integrated Genome Browser (IGB, v9.1.8) to visualize read density profiles. |
| | 10. SNP enrichment was calculated using GREGOR package (v1.4.0) |
| | 11. Heritability estimates were generated using LD score regression (LDSC) implemented in the ldsc package (v1.0.1, https://github.com/bulik/ldsc/) and SumHer implemented in LDAK software (version 5.2 , www.ldak.org). |
| | 12. Conditioning was performed using multi-trait-based conditional and joint analysis (mtCOJO) tool from GCTA pipeline (v1.94.1, https://yanglab.westlake.edu.cn/software/gcta/). |

13. TwoSampleMR (v0.5.6) R package was used for Mendelian randomisation (MR) analyses.
14. RIMBANET (version 26th June 2019, https://labs.icahn.mssm.edu/zhulab/?s=rimbanet) and Enrichr webserver (Version of 29th March 2021, https://maayanlab.cloud/Enrichr/) were used for Bayesian network queries for prioritized genes.
15. The code for the druggability analysis can be found in GitLab (https://cfinan.gitlab.io/biomisc/scripts/drug_lookups.html).

For manuscripts utilizing custom algorithms or software that are central to the research but not yet described in published literature, software must be made available to editors and reviewers. We strongly encourage code deposition in a community repository (e.g. GitHub). See the Nature Portfolio guidelines for submitting code & software for further information.

# Data

Policy information about availability of data

All manuscripts must include a data availability statement. This statement should provide the following information, where applicable:
- Accession codes, unique identifiers, or web links for publicly available datasets
- A description of any restrictions on data availability
- For clinical datasets or third party data, please ensure that the statement adheres to our policy

1. Gene reference names and coordinates were retrieved from GENCODE project through EBI FTP server. gencode.v38.annotation.gff3 and gencode.v38lift37.annotation.gff3 files were used.
2. eQTL data was retrieved from the Genotype Tissue Expression (GTEx) version 8 (https://gtexportal.org/home/datasets).
3. H3K27ac ChIP-Seq datasets (narrowpeaks beds) in any tissue were retrieved from ENCODE (https://www.encodeproject.org/).
4. Single nuclei ATAC-Seq peak files (bed format) from Human enhancer atlas (http://catlas.org/humanenhancer).
5. Open chromatin regions in healthy coronary arteries were generated from raw reads retrieved from Sequence Read Archive (SRR2378591, SRR2378592, SRR2378593).
6. Raw snATAC-Seq data in 25 adult tissues was retrieved from Gene Expression Omnibus (GSE184462).
7. Gene expression models for TWAS were retrieved from gusev lab website (http://gusevlab.org/projects/fusion/), based on GTEx data (v8 release).
8. Gene expression data from aorta artery, coronary artery, tibial artery, and cultured fibroblast was curated from the Genotype Tissue Expression (GTEx) version 8 (www.gtexportal.org/home/datasets).
9. Gene expression data from the mouse aorta was curated from the Hybrid Mouse Diversity Panel (HMDP).
10. Genes associated to mouse cardiovascular phenotypes (code MP:0005385) were retrieved from Mouse Genome informatics (www.informatics.jax.org).
11. Summary statistics were retrieved from http://www.cardiogramplusc4d.org/data-downloads/, http://ftp.ebi.ac.uk/pub/databases/gwas/summary_statistics/, https://www.megastroke.org/, http://www.nealelab.is/uk-biobank , https://diagram-consortium.org/downloads.html, or retrieved from authors as detailed in Supplementary table 17.
12. The set of genes encoding druggable targets was derived using ChEMBL v17, and further analyzed using ChEMBL v30 and the British National Formulary (BNF) (accessed 09/04/2021).
13. Summary statistics for SCAD association are available in GWAS catalog (GCP000522).

# Human research participants

Policy information about studies involving human research participants and Sex and Gender in Research.

| Reporting on sex and gender | Any findings in the present study only apply to sex and no indication about gender was collected and/or analyzed. Initial information about sex was collected by clinicians and the consistence with genotyping was assessed at the quality control step. Five samples were excluded due to a discrepancy between the clinical information and sex determined by genetic analysis. Sex stratified analyses were ran whenever possible. However, the low proportion of males in SCAD cohorts did not allow to perform a GWAS meta-analysis on males only. |
|---|---|
| Population characteristics | 1. DISCO, (France), Total (n): 313, Women (n,%): 285 (91), Age at SCAD (Median, Q1/Q3): 52.2, 44.55, 60, Age at study 2. inclusion (Median, Q1,Q3): 51, 44, 59, FMD (Yes, No, NA): 140, 152, 21. 3. 3C-Study, (France), Total (n): 1487, Women (n,%): 876 (58.9), Age at SCAD (Median, Q1/Q3): NR, 51, 44, 59: 74.36 ± 5.5 4. [65 - 94], 140, 152, 21: NR. 5. SCAD-UK Study I - Cases, (UK), Total (n): 383, Women (n,%): 361 (94.2), Age at SCAD (Median, Q1/Q3): 47, 41, 52, 74.36 ± 5.5 6. [65 - 94]: NA, NR: 104,108,171. 7. SCAD-UK Study I - Controls, (UK), Total (n): 1925, Women (n,%): 1815 (94.3), Age at SCAD (Median, Q1/Q3): NR, NA: 56,49,62, 104,108,171: . 8. SCAD-UK Study II - Cases, (UK), Total (n): 139, Women (n,%): 115 (82.7), Age at SCAD (Median, Q1/Q3): 49.0, 43, 54, 56,49,62: NA, : 20,71,48. 9. SCAD-UK Study II - Controls, (UK), Total (n): 815, Women (n,%): 665 (81.6), Age at SCAD (Median, Q1/Q3): NR, NA: 56, 48, 61, 20,71,48: . 10. Mayo Clinic Study - Cases, (US), Total (n): 506, Women (n,%): 484, Age at SCAD (Median, Q1/Q3): 46.6, 39, 53, 56, 48, 61: 46.6 ± 9.2, : 175, 140, 169. 11. Mayo Clinic Study - Controls, (US), Total (n): 1549, Women (n,%): 1477, Age at SCAD (Median, Q1/Q3): NR, 46.6 ± 9.2: 64 ± 14.5, 175, 140, 169: unknown. 12. CanSCAD/MGI Study - Cases, (Canada/US), Total (n): 357, Women (n,%): 315 (88.2%), Age at SCAD (Median, Q1/Q3): , 64 ± 14.5: 53, 46, 60, unknown: 149,123,85. 13. CanSCAD/MGI Study - Controls, (Canada/US), Total (n): 2125, Women (n,%): 1873 (88.1%), Age at SCAD (Median, Q1/Q3): NR, 53, 46, 60: 53, 46, 61, 149,123,85: NR. 14. DEFINE-SCAD Study - Cases, (US), Total (n): 42, Women (n,%): 41 (97.6%), Age at SCAD (Median, Q1/Q3): 45.5, 36, 50.25 15. 6 missing values, 53, 46, 61: 49, 41.5, 53.75, NR: 31, 10, 1. 16. DEFINE-SCAD Study - Controls, (US), Total (n): 153, Women (n,%): 153 (100%), Age at SCAD (Median, Q1/Q3): NA, 49, 41.5, 53.75: 50 (43-58), 31, 10, 1: NR. |

17. VCCRI Study I - Cases, (Australia), Total (n): 88, Women (n,%): 80, 90.9%, Age at SCAD (Median, Q1/Q3): 44, 39, 52, 50 (43-58): 50, 44, 59, NR: 14, 32, 42.
18. VCCRI Study I - Controls, (Australia), Total (n): 1127, Women (n,%): 672, 59.6%, Age at SCAD (Median, Q1/Q3): NA, 50, 44, 59: all >70 years old, 14, 32, 42: NR.
19. VCCRI Study II - Cases, (Australia), Total (n): 85, Women (n,%): 83, 97.6%, Age at SCAD (Median, Q1/Q3): 49, 43, 56, all >70 years old: 52, 48, 60, NR: 10, 22, 53.
20. VCCRI Study II - Controls, (Australia), Total (n): 111, Women (n,%): 46, 41.4%, Age at SCAD (Median, Q1/Q3): NA, 52, 48, 60: 61, 52, 67, 10, 22, 53: NR.

| Recruitment | |
|---|---|

Altogether, the meta-analysis included participants of European ancestry from eight studies: DISCO-3C, SCAD-UK I, SCAD-UK II, Mayo Clinic, DEFINE-SCAD, CanSCAD/MGI, VCCRI I and VCCRI II. SCAD patients presented similar clinical characteristics and homogeneous diagnosis, exclusion and inclusion criteria. Controls were selected from local population-based studies or clinical studies led in the same centers. In the second case studies, SCAD or related vascular diseases were exclusion criteria. The rare presence of males in SCAD cohorts may be partly due to a lack of diagnosis in this population, considering that women are considered to be more at risk of SCAD. The limited presence of non-European cases and controls, likely related at least in part to socio-economic factors, prevents the analysis of these populations.
1. DISCO, (France).
1) Method of recruitment: Clinical based.
2) Inclusion criteria: age> 18, retrospective with a diagnostic of SCAD made from 2010, or prospective at the time of hospitalization during which the diagnosis of SCAD was made.
3) Exclusion criteria: Age <18; atherosclerotic ischemic disease; iatrogenic hematoma.
2. 3C-Study, (France).
1) Method of recruitment: Population based.
2) Inclusion criteria: Geographic sampling.
3) Exclusion criteria: Age < 65y.
3. SCAD-UK Study I - Cases, (UK).
1) Method of recruitment: Clinical based.
2) Inclusion criteria: SCAD confirmed on invasive angiography.
3) Exclusion criteria: Atherosclerotic dissection, iatrogenic dissection.
4. SCAD-UK Study I - Controls, (UK).
1) Method of recruitment: Population based.
2) Inclusion criteria: None.
3) Exclusion criteria: None.
5. SCAD-UK Study II - Cases, (UK).
1) Method of recruitment: Clinical based.
2) Inclusion criteria: SCAD confirmed on invasive angiography.
3) Exclusion criteria: Atherosclerotic dissection, iatrogenic dissection.
6. SCAD-UK Study II - Controls, (UK).
1) Method of recruitment: Population based.
2) Inclusion criteria: None.
3) Exclusion criteria: None.
7. Mayo Clinic Study - Cases, (US).
1) Method of recruitment: Clinical based.
2) Inclusion criteria: SCAD confirmed by angiogram.
3) Exclusion criteria: Diagnosis of connective tissue disorder or aortopathy; iatrogenic.
8. Mayo Clinic Study - Controls, (US).
1) Method of recruitment: Healthy volunteers.
2) Inclusion criteria: No reported SCAD.
3) Exclusion criteria: Diagnosis of atherosclerotic coronary artery disease, acute myocardial infarction, FMD, arterial aneurysm or dissection, cerebral infarction, Marfan syndrome, Ehlers-Danlos syndrome.
9. CanSCAD/MGI Study - Cases, (Canada/US).
1) Method of recruitment: Clinical based.
2) Inclusion criteria: SCAD diagnosis was confirmed on coronary angiography by the UBC core laboratory research team, and categorized according to previously established Saw classification.
3) Exclusion criteria: Angiogram unavailable or did not appear to be SCAD; from N=502, only Canadian samples consistent with 1000G non-Finish European ancestry (+/- 6 SD of PC1 and PC2) were retained for analysis. .
10. CanSCAD/MGI Study - Controls, (Canada/US).
1) Method of recruitment: Population based.
2) Inclusion criteria: Age, Sex, PC (PC1-PC3) matched controls.
3) Exclusion criteria: Of 13,756 MGI samples eligible for the study after exclusion of vascular or connective tissue diagnoses, and matching for age, sex and ancestry (based upon genetic PC's) 2,125 matched MGI controls were retained for analysis.
11. DEFINE-SCAD Study - Cases, (US).
1) Method of recruitment: Clinical based.
2) Inclusion criteria: SCAD confirmed on invasive angiography.
3) Exclusion criteria: Age < 18, diagnosis of connective tissue disorder or aortopathy; iatrogenic. Any diagnosis of other major diseases.
12. DEFINE-SCAD Study - Controls, (US).
1) Method of recruitment: Clinical based.
2) Inclusion criteria: Vascular disease excluded on history and physical exam. Also matched to SCAD cases by age, BMI, sex.
3) Exclusion criteria: Any diagnosis of vascular disease and other major diseases.
13. VCCRI Study I - Cases, (Australia)
1) Method of recruitment: Clinical based.
2) Inclusion criteria: SCAD confirmed by angiogram.
3) Exclusion criteria: Angiogram unavailable or did not appear to be SCAD.

14. VCCRI Study I - Controls, (Australia).
1) Method of recruitment: Population based.
2) Inclusion criteria: No reported SCAD.
3) Exclusion criteria: No reported history of cancer, cardiovascular disease or neurodegenerative diseases before 70 years old.
15. VCCRI Study II - Cases, (Australia).
1) Method of recruitment: Clinical based.
2) Inclusion criteria: SCAD confirmed by angiogram.
3) Exclusion criteria: Angiogram unavailable or did not appear to be SCAD.
16. VCCRI Study II - Controls, (Australia).
1) Method of recruitment: Clinical Based.
2) Inclusion criteria: No Reported SCAD.
3) Exclusion criteria: Related to other sample.

| Ethics oversight | 1. DISCO, (France): Clinical Trials ID: NCT02799186, regional committee CPP (comité de protection des personnes) Sud-Est 6 2016 AU-1258 |
|---|---|

2. 3C-Study, (France): «comité consultatif de protection des personnes dans la recherche biomédicale» Bicêtre Hôpital Bicêtre n°99-28 CCPPRB approved 10/06/99, 11/03/2003 and 17/03/2006.
3. SCAD-UK Study (UK): The UK SCAD study (ISRCTN42661582) was approved by the UK National Research Ethics Service (14/EM/0056) and the UK Health Research Authority.
4. Mayo Clinic Study (US): Mayo Clinic Institutional Review Board (NCT01429727; NCT01427179).
5. CanSCAD/MGI Study (Canada/US): Research ethics board approvals were obtained at each site of SCAD patient inclusion, and all patients provided informed consent for participation. IRB approval: HUM00113268, SCAD Registry and Research. IRB approval: HUM00112101, genetic analysis of arterial dysplasia and remodeling (MGI/AOS)
6. DEFINE-SCAD Study - Cases, (US): DEFINE study was approved by the Human Research Ethics Committee of the Icahn School of Medicine at Mount Sinai (Study ID: HS#13-00575/GCO#13–1118 and is registered with ClinicalTrials.gov Identifier: NCT01967511.
7. VCCRI Study I, (Australia): St. Vincent's Hospital Human Research Ethics Committee (HREC/16/SVH/338, protocol number SVH 16/245)
8. VCCRI Study II - Cases, (Australia): St. Vincent's Hospital Human Research Ethics Committee (HREC/17/SVH/315)

Note that full information on the approval of the study protocol must also be provided in the manuscript.

# Field-specific reporting

Please select the one below that is the best fit for your research. If you are not sure, read the appropriate sections before making your selection.

☒ Life sciences          ☐ Behavioural & social sciences          ☐ Ecological, evolutionary & environmental sciences

For a reference copy of the document with all sections, see nature.com/documents/nr-reporting-summary-flat.pdf

# Life sciences study design

All studies must disclose on these points even when the disclosure is negative.

| Sample size | Recruitment of SCAD patients was the main limiting factor for sample size of GWAS case-control studies and by far represents the more thorough genetic investigation in SCAD. In 7 out of 8 case-control studies, the number of controls was picked to have a minimum 2.5x ratio over SCAD cases, in order to allow a robust measure of variant frequency, and maximum ratio of 15x the number of SCAD cases, to avoid artifical inflation of effective sample size. Number of controls in VCCRI arm II study was more limited due to constraints of clinical recruitment, but was set to outnumber SCAD patients in the study. |
|---|---|
| Data exclusions | 1. Several datasets were excluded on the basis of classical quality control criteria. The detail of excluded datasets per study and per criterion is given in Supplementary table 13 and in the Methods and Supplementary Methods sections. In brief, 49 samples were excluded from CanSCAD study because they wer involved in another case-control study, 39 samples were excluded on the basis of heterozigocity/call rate assessment, 10 samples were excluded because of a high relatedness with other samples in the same study, 219 samples were excluded based on corrected diagnosis (non-SCAD), 2 samples were excluded due to a diagnosis of another genetic syndrome and 5 samples were excluded because sex determined by genotyping or sequencing did not match clinical data. All these criteria were included in study design. In addition, the analysis was restricted to European samples based on principal component analysis, as the number of non-European samples in control cohorts was insufficient.<br><br>2. Prior to meta-analysis, we removed single nucleotide polymorphisms (SNPs) with low minor allele frequencies (MAF<0.01), low imputation quality (r2 < 0.8), and deviations from Hardy-Weinberg equilibrium (P<10−5). |
| Replication | Here we only considered as risk loci for SCAD those that provided several associated SNPs with SCAD consistently across the 8 case control studies, with the same direction of effects, and no evidence for heterogeneity between studies We consider these criteria as evidence of replication across studies, given that we used data from 6 independant recruitment centers from 5 countries. To maximise power, we explored results from the meta-analysis. Details about the association of each genome-wide associated locus are presented in Suppeelementary table 2 |
| Randomization | Individuals were allocated to patients and control groups through clinical recruitment and could not be randomized. To adjust for population |

| | |
|---|---|
| Randomization | covariates, genetic models were adjusted for population structure using the first five principal components, sex (except in the women-only analyses) and study specific genomic control. |
| Blinding | The investigators could not be blinded during data collection and analysis, as case and control cohorts originated from different protocols and/or required a thorough clinical investigation. |

# Reporting for specific materials, systems and methods

We require information from authors about some types of materials, experimental systems and methods used in many studies. Here, indicate whether each material, system or method listed is relevant to your study. If you are not sure if a list item applies to your research, read the appropriate section before selecting a response.

## Materials & experimental systems

| n/a | Involved in the study |
|---|---|
| ☒ | Antibodies |
| ☒ | Eukaryotic cell lines |
| ☒ | Palaeontology and archaeology |
| ☒ | Animals and other organisms |
| ☒ | Clinical data |
| ☒ | Dual use research of concern |

## Methods

| n/a | Involved in the study |
|---|---|
| ☒ | ChIP-seq |
| ☒ | Flow cytometry |
| ☒ | MRI-based neuroimaging |

