## [Peer Review File · Nature Genetics]

Peer Review Information

Manuscript Title: Genome-wide association meta-analysis of spontaneous coronary artery dissection identifies risk variants and genes related to artery integrity and tissue-mediated coagulation

Corresponding author name(s): Dr Nabila Bouatia-Naji, David Adlam

Reviewer Comments & Decisions:

Decision Letter, First revision:

17th November 2022

Dear Nabila,

Your Article "Genome-wide association meta-analysis of spontaneous coronary artery dissection reveals common variants and genes related to artery integrity and tissue-mediated coagulation" has been seen by three referees. You will see from their comments below that, while they find your work of potential interest, they have raised several relevant points. We are interested in the possibility of publishing your study in Nature Genetics, but we would like to consider your response to these points in the form of a revised manuscript before we make a final decision on publication.

To guide the scope of the revisions, the editors discuss the referee reports in detail within the team, including with the chief editor, with a view to identifying key priorities that should be addressed in revision, and sometimes overruling referee requests that are deemed beyond the scope of the current study. In this case, we ask that you address all technical queries related to analyses and their interpretation, revising the presentation of findings where needed, and extend the analyses and discussion along the lines suggested by the referees. We hope you will find this prioritized set of referee points to be useful when revising your study. Please do not hesitate to get in touch if you would like to discuss these issues further.

We therefore invite you to revise your manuscript taking into account all reviewer and editor comments. Please highlight all changes in the manuscript text file. At this stage we will need you to upload a copy of the manuscript in MS Word .docx or similar editable format.

*2) If you have not done so already, please begin to revise your manuscript so that it conforms to our Article format instructions, available [here](http://www.nature.com/ng/authors/article_types/index.html). Refer also to any guidelines provided in this letter.

[redacted]

We hope to receive your revised manuscript within 8-12 weeks. If you cannot send it within this time, please let us know.

Nature Genetics is committed to improving transparency in authorship. As part of our efforts in this direction, we are now requesting that all authors identified as 'corresponding author' on published papers create and link their Open Researcher and Contributor Identifier (ORCID) with their account on the Manuscript Tracking System (MTS), prior to acceptance. ORCID helps the scientific community achieve unambiguous attribution of all scholarly contributions. You can create and link your ORCID from the home page of the MTS by clicking on 'Modify my Springer Nature account'. For more information, please visit www.springernature.com/orcid.

Sincerely,
Kyle

Kyle Vogan, PhD

Senior Editor
Nature Genetics
<https://orcid.org/0000-0001-9565-9665>

Referee expertise:

Referee #1: Genetics, cardiovascular diseases, clinical translation

Referee #2: Genetics, cardiovascular diseases, functional genomics

Referee #3: Genetics, cardiovascular diseases, clinical translation

Reviewers' Comments:

Reviewer #1:
Remarks to the Author:

This is the first large GWAS meta-analysis for SCAD, an important and understudied cause of acute myocardial infarction in the absence of obstructive coronary artery disease and predominately affecting younger female individuals.

I applaud the successful recruitment of cases and controls from 8 sites on 3 continents.

The authors state that 17 genome-wide significant loci were identified of which 12 were new. Table 1 lists a total of 18 loci (accounting for independent signals on COL4A1 and COL4A2). Suggest state the same number of loci throughout.

Figure 1 increases the number of potential causal genes to 27 by including up to 4 genes for some loci based on several functional annotation methodologies, here described in detail. Could a single gene be prioritized for each locus?

Supp Fig 8. All 27 genes are included in the Bayesian networks In Supp Fig 8. I am not convinced that the data warrant inclusion of the separate clusters driven by JUN, CTSS and TIMP3.

Although most of the new findings are sufficiently robust to include in the manuscript for further investigation, I suggest removal of the rs137507 and rs5973204 signals:

- Supportive data for the rs137507 locus (SYN3 or TIMP3) is weak and the p value barely significant.
- SRSF2P1 (a pseudogene) and TMEM47 are the only genes near rs5973204 on the X chr and lack molecular, clinical, or experimental data.

Although PHACTR1 is among the candidate genes previously identified for SCAD (and coronary calcification), a detailed in vivo analysis of PHACTR1 in the vasculature using 3 separate knockout models revealed no effects, suggesting the EDN1 located 600 kb upstream of PHACTR1 might be the causal gene (PMID: 35387477). EDN1 is linked to vasoconstriction and SMC proliferation, seemingly more relevant to SCAD. The authors should discuss this point.

Of interest, the authors report that several of the SCAD associated variants exhibit opposite effects for CAD. This is a provocative finding but fits with clinical data showing no association between coronary atherosclerosis and SCAD and an earlier report on the PHACTR1/EDN1 locus (PMID 30621952).

Minor Points

Supp Fig 1: change US to US/CDA or NAm

Supp Fig 10: I am not sure why several very closely related traits are included, e.g. Hb and HCT, TC and LDL

P 7 line 257: change "several shared loci" to "one shared locus" rs34370185 (possibly two with rs11838776)

Reviewer #2:

Remarks to the Author:

Adlam and colleagues here report genome-wide association study (GWAS) results for spontaneous coronary artery dissection (SCDA), a severe and relatively uncommon cardiovascular disease that mostly affects women. The authors increased the sample size – now totalling 1917 cases and 9292 controls – and identified 17 risk loci, including 12 novel genomic regions. Then, they applied a battery of in silico analysis strategies to prioritize variants, genes and cell-types. They also calculated heritability, computed genetic correlations with other diseases and traits, and attempted to dissect causality using Mendelian randomization techniques. The most remarkable result is the consistent inverse effect of associated alleles on SCAD and coronary artery disease (CAD) risk. The main strengths include: (1) a focus on an uncommon cardiovascular disease for which there is little pathophysiological insights and (2) a well-written manuscript describing a series of standard in silico analyses. In terms of limitations, I noted: (1) a lack of replication, (2) no attempt to functionally characterize the genetic discoveries, and (3) no explanation as to why SCAD affects mostly women. I have the following comments:

1. While I understand that increasing sample size for SCAD is challenging, it remains that most loci have not been replicated and could be false positive associations. The authors should acknowledge this limitation in the Discussion.
2. SCAD is more prevalent in women, whereas CAD is more frequent in men. And the alleles that increase SCAD risk reduce CAD risk. One potential trivial explanation to this observation is that the alleles are associated with sex (as opposed to SCAD and CAD). I know this sounds very unlikely, but could the authors check that alleles at the identified loci have similar frequencies in men and women?
3. What is known about SCAD in non-European-ancestry individuals? Similar prevalence? Can you comment on this in the Discussion?
4. Figure 4A. Are all variants in the 95% credible sets considered in this enrichment analysis? For these analyses, I understand that the matched sets are generated using size and chromosome. But is that sufficient to generate a true matched null set? Shouldn't you consider other important parameters

that may define regulatory sequences such as content (%GC) or proximity to genes? Would you get the same enrichment results if you were to use the more standard approach, which is to match variants (as opposed to peaks)?

5. Figure 4B. What is the significance threshold? Also, when there is no symbol, is it correct to assume that there is no GTEx models? This should be clarified in the figure legend.

6. How were MR instruments selected? What steps did you take to ensure that the MR assumptions are respected?

Reviewer #3:

Remarks to the Author:

Adlam and colleagues present the results of a genome-wide association meta-analysis of spontaneous coronary artery dissection (SCAD) totaling 1,917 SCAD cases of European ancestry and 9,292 ancestry matched controls free of SCAD from 8 prior case-control studies. A total of 17 risk loci were identified, 12 of which are novel at genome-wide significance in this study (4 of which were identified as having a suggestive level of association in the prior study of Saw et al., Nature Comm 2020). The authors confirm that SCAD is heritable and estimate that the identified loci account for ~24% of SNP-based heritability (similar to the prior estimate in Saw et al., Nature Comm 2020). The authors perform in silico based annotation of probable causal genes and identify a shared genetic basis between SCAD and arterial disease, confirming the opposite genetic effect of variants for SCAD and atherosclerotic coronary artery disease (first identified in Saw et al, Nature Comm 2020).

I complement the authors on a well-executed GWAS meta-analysis. The methods appear to be sound and conducted appropriately.

The identification of F3 as a risk locus is interesting and the authors propose a mechanism of decreased F3 expression as driving risk. Did the authors look at loci encoding other components of the coagulation cascade to see if there are sub genome-wide significant results in other loci? F3 forms a complex with factor VIIa, for example, so one might expect to see some signal in the F7 locus. Similarly, did the authors consider performing a PheWAS for the F3 variant to see what other bleeding disorders (or other phenotypes) are co-associated?

The authors suggest in a few places that this work highlights potential therapeutic strategies. Have the authors systematically characterized the loci to determine which encode druggable targets?

Many of the major results and conclusions seem to be highlighted in prior work (with the exception of novel loci identified here) as summarized recently by Weldy et al., Circulation Genomics and Precision Medicine 2022. The authors might emphasize the truly novel findings here beyond what has been reported in prior publications.

A trivial point but the chromosome 15 designation is missing from the THSD4 locus in Table 1.

Author Rebuttal, first revision:

Referee expertise:

Referee #1: Genetics, cardiovascular diseases, clinical translation

Referee #2: Genetics, cardiovascular diseases, functional genomics

Referee #3: Genetics, cardiovascular diseases, clinical translation

Reviewer #1:**Remarks to the Author:**

This is the first large GWAS meta-analysis for SCAD, an important and understudied cause of acute myocardial infarction in the absence of obstructive coronary artery disease and predominately affecting younger female individuals.

I applaud the successful recruitment of cases and controls from 8 sites on 3 continents.

1. The authors state that 17 genome-wide significant loci were identified of which 12 were new. Table 1 lists a total of 18 loci (accounting for independent signals on COL4A1 and COL4A2). Suggest state the same number of loci throughout.

We thank the reviewer for their suggestion to clarify the number of significant loci. We initially reported 17 significant loci, where we defined each locus by a lead SNP and its surrounding genomic region (+/- 500kbp). One locus (*COL4A1/COL4A2*) included two independent signals according to conditional regression analyses reported in Supplementary Figure 3. To clarify this point, we have now updated Table 1 by including a column labeled locus which help identify loci and specify that COL4A1 and COL4A2 are two independent signals located in the same locus. Following the suggestion to re-evaluate chromosome X signal, we have a total number of 16 loci, including 11 new (See response to Point 4 for details).

2. Figure 1 increases the number of potential causal genes to 27 by including up to 4 genes for some loci based on several functional annotation methodologies, here described in detail. Could a single gene be prioritized for each locus?

To prioritize target genes in SCAD loci, we applied a multi-sources annotation strategy that led, in some loci, to more than 1 candidate gene in the same locus. Following the reviewer comment, and in the attempt to narrow down the genes per loci, we decided to give more weight to genes supported by high posterior probability of colocalization between the GWAS signal and eQTL association in tissues of interest to SCAD, and TWAS hits. Other important criteria, mainly biological relevance to cardiovascular disease (e.g. Mouse CV phenotype and Human CVD) and distance to association signal were applied as secondary

criteria and to help prioritize one gene in some loci. We have now a list of 21 candidate genes belonging to the 17 loci.

We note that this new strategy still prioritized 2 genes (*ADAMTSL4* and *ECM1*) on Chr1q21 that showed identical scores, and 3 genes (*MRPS6*, *SLC5A3* and *KCNE2*) on Chr21q22. We found that in the case of the Chr21q22 locus, all three genes present compelling arguments supporting each of them as a potential biological target in the case of arterial dissection. In this case, only experimentally-based biological exploration, which we believe is beyond the scope of this study, would be able to identify the target gene in the case of SCAD, under the hypothesis of one unique target gene in the locus.

The results section was updated on Page 5, to reflect the changes in the prioritization strategy and results.

We note that we took the opportunity of the revision process to update our TWAS analyses using a more recent version of GTEx dataset (v8 release). Figures 1 and 2, Supplementary Figure 8, Supplementary Table 9 and main text were updated accordingly.

3. Supp Fig 8. All 27 genes are included in the Bayesian networks In Supp Fig 8. I am not convinced that the data warrant inclusion of the separate clusters driven by JUN, CTSS and TIMP3.

Following the update of the list of prioritized genes, we have re-run the Bayesian network analysis that we present now as Supp. Figure 9. In this new set of 20 genes, *TIMP3* has clustered in the main network “Extracellular Matrix Organization” that includes *ADAMTSL4*, *LRP1*, *COL4A1*, with connections with sub-networks of F3 for instance. This clustering is consistent with the biological function of *TIMP3* as an inhibitor of matrix metalloproteinases with interaction domains in *ADAMTS* proteins and *LRP1*, and high relevance to vascular biology (reviewed in PMID: 32612540).

4. Although most of the new findings are sufficiently robust to include in the manuscript for further investigation, I suggest removal of the rs137507 and rs5973204 signals:
 - Supportive data for the rs137507 locus (*SYN3* or *TIMP3*) is weak and the p value barely significant.
 - *SRSF2P1* (a pseudogene) and *TMEM47* are the only genes near rs5973204 on the X chr and lack molecular, clinical, or experimental data.

To compensate for the relatively limited power in our meta-analysis, we only considered for follow-up and, functionally annotated robust association signals, according to stringent quality control criteria (See details provided for Reviewer #2). In the revised manuscript, we have added a Supplementary Table where we report the association results per study to allow the appreciation of odds ratios estimates and p-values in individual studies, which supports solid statistical evidence for association with SCAD. Nonetheless, in the specific case of the 2 loci highlighted by this Reviewer, we agree that the association signal observed on the X chromosome present some weaknesses. Despite going through all QC filters, two studies did not cover the 2 main SNPs that contribute to this signal (R1 Figure): Mayo Clinic (506 cases and 1549 controls), and SCAD-UKII (143 cases and 815 controls).

R-Figure1. LocusZoom generated from the SCAD GWAS meta-analysis on ChrX association signal.

Given the lower level of significance of the other SNPs from the same LD block, we agree that further replication is needed to robustly declare this locus as GWAS significant and agree to remove it from our current list. In addition, the function of potential target gene, TMEM47, is less clearly related to other SCAD associated genes. We found no evidence for an eQTL signal in this locus in arteries or fibroblasts, despite a high level of expression in arteries of TMEM47 reported in GTEx.

As for the Chr22 signal, we found several lines of evidence to support the relevance of TIMP3 locus and its link to the risk of SCAD that we tend to consider as robust. First, the association signal is backed by a

large number of sub-GWAS associated SNPs from the same LD block, that show consistent association across individual cases control studies (See Supplementary Table 2, R-Figure 2).

R-Figure2: LocusZoom generated from the SCAD GWAS meta-analysis on Chr22 association signal near TIMP3.

Second, many of these SNPs overlap with open chromatin regions in vascular smooth muscle cells, which supports a high potential for functionality of this genomic region. Third, this locus was recently identified as a newly associated with coronary artery disease in a recently published multiethnic GWAS (PMID: 35915156). Fourth, the prioritized gene TIMP3 encodes the tissue inhibitor of metalloproteinase 3, a key protein in the stability and normal function of the extracellular matrix of arteries. Fifth, there is solid evidence through documented functional interaction in the context of arterial diseases between TIMP3 and several proteins the genes of which are prioritized in SCAD loci, including LRP1 (PMID: 23166318), HTRA1 (PMID: 29725820), and type IV Collagen (PMID: 12798442). We have updated the discussion with these relevant features of TIMP3 candidacy in SCAD pathophysiology. (Page 9, Paragraph 2).

5. Although PHACTR1 is among the candidate genes previously identified for SCAD (and coronary calcification), a detailed in vivo analysis of PHACTR1 in the vasculature using 3 separate knockout models revealed no effects, suggesting the EDN1 located 600 kb upstream of PHACTR1 might be the causal gene (PMID: 35387477). EDN1 is linked to vasoconstriction and SMC proliferation, seemingly more relevant to SCAD. The authors should discuss this point.

PHACTR1 is a pleiotropic risk locus for a large number of vascular diseases, including SCAD, that we have reported in our previous studies (PMID: 30621952, 32887874, 32887874). The application of our prioritization strategy did not identify EDN1 as a prioritized gene in this locus, similarly to previous work on coronary artery disease (PMID: 34961328) and fibromuscular dysplasia (PMID: 34654805) and is in keeping with previous work on gene expression in iPSCs (PMID: 30354304). We agree this recent mouse study mentioned by this Reviewer provides interesting data, although this work did not explore measures of arterial distensibility/compliance which we have demonstrated in humans to be impacted by genotype at this locus (PMID: 35653516). Given the debate about the biological mechanism potentially at play, we decided to focus the results and discussion on genes from novel loci described for the first time for SCAD, as recommended by Reviewer #3.

6. Of interest, the authors report that several of the SCAD associated variants exhibit opposite effects for CAD. This is a provocative finding but fits with clinical data showing no association between coronary atherosclerosis and SCAD and an earlier report on the PHACTR1/EDN1 locus (PMID 30621952).

We agree with the reviewer and have added a short statement to the final paragraph of the discussion to emphasize this point (Page 9).

Minor Points

1. Supp Fig 1: change US to US/CDA or NAm
We have updated this Supplementary Figure accordingly.

2. Supp Fig 10: I am not sure why several very closely related traits are included, e.g. Hb and HCT, TC and LDL
We have now updated all figures and supplementary tables with only independent traits and applied multiple testing correction accordingly.

3. P 7 line 257: change “several shared loci” to “one shared locus” rs34370185 (possibly two with rs11838776)
This sentence was changed in the revised manuscript as requested.

Reviewer #2:

Remarks to the Author:

Adlam and colleagues here report genome-wide association study (GWAS) results for spontaneous coronary artery dissection (SCAD), a severe and relatively uncommon cardiovascular disease that mostly affects women. The authors increased the sample size – now totalling 1917 cases and 9292 controls – and identified 17 risk loci, including 12 novel genomic regions. Then, they applied a battery of in silico analysis strategies to prioritize variants, genes and cell-types. They also calculated heritability, computed genetic correlations with other diseases and traits, and attempted to dissect causality using Mendelian randomization techniques. The most remarkable result is the consistent inverse effect of associated alleles on SCAD and coronary artery disease (CAD) risk. The main strengths include: (1) a focus on an uncommon cardiovascular disease for which there is little pathophysiological insights and (2) a well-written manuscript describing a series of standard in silico analyses.

In terms of limitations, I noted: (1) a lack of replication, (2) no attempt to functionally characterize the genetic discoveries, and (3) no explanation as to why SCAD affects mostly women. I have the following comments:

1. While I understand that increasing sample size for SCAD is challenging, it remains that most loci have not been replicated and could be false positive associations. The authors should acknowledge this limitation in the Discussion.

We agree that replication in independent studies is key to confidently declare genetic association results from GWAS. As acknowledged by this Reviewer, SCAD diagnosis is challenging and the current study contained probably all existing cohorts with DNA available for GWAS analyses. Controlling for false positives was a key step of our quality control. Prior to the meta-analysis, each study applied stringent criteria (excluded SNPs with $MAF < 0.01$, low imputation quality ($r^2 < 0.8$), and those showing deviations from Hardy-Weinberg equilibrium ($P < 10^{-5}$). Our meta-analysis of GWAS was generated from 8 case controls studies, with patients and genetic data generated from 6 different research centers (France, UK, UBC/MGI, Mayo Clinic and VCCRI). Here, we only considered as risk loci for SCAD those that provided several associated SNPs with SCAD consistently across the 8 case control studies, with the same direction of effects, and no evidence for heterogeneity between cohorts. We believe this strategy to be stringent and robust against false positive associations, although we acknowledge, given the limited global sample size, that future case control studies will be necessary to provide further evidence for associations with SCAD.

To allow the future readers of our manuscript and the Reviewers to assess the consistency of the associations of our association results, we now provide a Supplementary Table 2 where we provide the effects sizes and p-values for each contributing case control study. As a limitation, we mention now in the discussion that our results will benefit from future validation in larger GWAS settings.

2. SCAD is more prevalent in women, whereas CAD is more frequent in men. And the alleles that increase SCAD risk reduce CAD risk. One potential trivial explanation to this observation is that the alleles are associated with sex (as opposed to SCAD and CAD). I know this sounds very unlikely, but could the authors check that alleles at the identified loci have similar frequencies in men and women?

We checked the effect allele frequencies (EAF) for all lead SNPs in the reported loci in the overall sample from meta-analyses for SCAD and CAD to compare them by sex and found no evidence for EAF differences according to sex (R-Table1). Estimates in SCAD males are missing in this table due to the very limited numbers available.

Locus	Annotated Genes	CHR:POS	rsID	REF	ALT	SCAD_bothsex	CAD_bothsex	SCAD_female	CAD_female	CAD_male
						meta_EAF	meta_EAF	meta_EAF	meta_EAF	meta_EAF
1	FGGY-DT	1:59656909	rs34370185	G	T	0.29	0.28	0.29	0.28	0.28
2	F3	1:95050472	rs1146473	T	C	0.19	0.19	0.19	0.19	0.19
3	ECM1/ADAMTSL4	1:150504062	rs4970935	C	T	0.72	0.73	0.72	0.73	0.73
4	AFAP1	4:7774352	rs6828005	G	A	0.55	0.53	0.55	0.53	0.53
5	ZNF827	4:146788035	rs1507928	T	C	0.48	0.48	0.48	0.48	0.48
6	ITGA1	5:52155642	rs73102285	A	G	0.27	0.25	0.27	0.25	0.25
7	PHACTR1	6:12903957	rs9349379	A	G	0.38	0.41	0.38	0.41	0.41
8	HTRA1	10:124259062	rs2736923	G	A	0.89	-	0.89	-	-
9	SES3	11:95308854	rs11021221	T	A	0.17	0.17	0.17	0.17	0.17
10	LRP1	12:57527283	rs11172113	T	C	0.38	0.41	0.38	0.41	0.41
11	ATP2B1	12:89978233	rs1689040	C	T	0.41	0.41	0.41	0.41	0.41
12	COL4A1	13:110838236	rs7326444	G	A	0.36	0.36	0.35	0.36	0.36
12	COL4A2	13:111040681	rs11838776	G	A	0.27	0.28	0.27	0.28	0.28
13	FBN1	15:48763754	rs7174973	A	G	0.11	0.10	0.12	0.10	0.10
14	THSD4	15:71628370	rs10851839	T	A	0.68	0.66	0.68	0.66	0.66
15	MRPS6/SLC5A3/KCNE2	21:35593827	rs28451064	G	A	0.12	0.13	0.12	0.13	0.13
16	TIMP3	22:33282971	rs137507	T	C	0.89	0.90	0.89	0.90	0.90

R-Table1. Effect allele frequencies for lead SNPs at SCAD loci in the whole sample and females only of SCAD meta-analysis, and whole sample, females and males from the latest CARDIoGRAMplusC4D Consortium meta-analysis.

We also looked up all lead SNPs in the GWAS association with sex from the UK Biobank accessed through (<http://www.nealelab.is/uk-biobank>) and found no evidence for significant association with sex.

Locus	Candidate Genes	CHROM	POS	rsID	REF	ALT	variant	beta	se	tstat	pval
1	FGGY-DT	1	59656909	rs34370185	G	T	1:59656909:G:T	-1.32E-03	1.30E-03	-1.02E+00	3.09E-01
2	F3	1	95050472	rs1146473	T	C	1:95050472:T:C	1.84E-03	1.50E-03	1.23E+00	2.19E-01
3	ECM1/ADAMTSL4	1	150504062	rs4970935	C	T	1:150504062:C:T	1.04E-03	1.34E-03	7.77E-01	4.37E-01
4	AFAP1	4	7774352	rs6828005	G	A	4:7774352:G:A	-4.01E-04	1.18E-03	-3.41E-01	7.33E-01
5	ZNF827	4	146788035	rs1507928	T	C	4:146788035:T:C	-1.37E-03	1.18E-03	-1.17E+00	2.42E-01
6	ITGA1	5	52155642	rs73102285	A	G	5:52155642:A:G	-2.09E-03	1.36E-03	-1.54E+00	1.24E-01
7	PHACTR1	6	12903957	rs9349379	A	G	6:12903957:A:G	-4.43E-04	1.19E-03	-3.71E-01	7.11E-01
8	HTRA1	10	124259062	rs2736923	G	A	10:124259062:G:A	-4.57E-04	1.92E-03	-2.37E-01	8.12E-01
9	SES3	11	95308854	rs11021221	T	A	11:95308854:T:A	1.01E-04	1.56E-03	6.49E-02	9.48E-01
10	LRP1	12	57527283	rs11172113	T	C	12:57527283:T:C	1.03E-03	1.19E-03	8.68E-01	3.85E-01
11	ATP2B1	12	89978233	rs1689040	C	T	12:89978233:C:T	3.30E-04	1.20E-03	2.76E-01	7.83E-01
12	COL4A1	13	110838236	rs7326444	G	A	13:110838236:G:A	-2.59E-03	1.22E-03	-2.11E+00	3.48E-02
12	COL4A2	13	111040681	rs11838776	G	A	13:111040681:G:A	-2.81E-05	1.32E-03	-2.12E-02	9.83E-01
13	FBN1	15	48763754	rs7174973	A	G	15:48763754:A:G	-2.11E-03	1.98E-03	-1.07E+00	2.85E-01
14	THSD4	15	71628370	rs10851839	T	A	15:71628370:T:A	6.74E-04	1.25E-03	5.40E-01	5.90E-01
15	MRPS6/SLC5A3/KCNE2	21	35593827	rs28451064	G	A	21:35593827:G:A	-3.04E-03	1.77E-03	-1.72E+00	8.57E-02
16	TIMP3	22	33282971	rs137507	T	C	22:33282971:T:C	-1.20E-03	1.96E-03	-6.11E-01	5.41E-01

R-Table2. Look-up for the association of lead SNPs at SCAD loci with sex.

3. What is known about SCAD in non-European-ancestry individuals? Similar prevalence? Can you comment on this in the Discussion?

Whilst most large observational series in SCAD have been dominated by patients of white European-ancestry, the most ethnically diverse population study (16% Black, 45% Hispanic) showed similar disease prevalence by ethnicity (PMID: 31084345). Series are described worldwide, for example in Japan (PMID: 26820364), South Korea (PMID: 31311261) and China (PMID: 36647158). To date no study has suggested a different prevalence or disease characteristics for SCAD in non-European-ancestry individuals although data remain relatively limited. We have added a comment on this to the Discussion.

4. Figure 4A. Are all variants in the 95% credible sets considered in this enrichment analysis? For these analyses, I understand that the matched sets are generated using size and chromosome. But is that sufficient to generate a true matched null set? Shouldn't you consider other important parameters that may define regulatory sequences such as content (%GC) or proximity to genes? Would you get the same enrichment results if you were to use the more standard approach, which is to match variants (as opposed to peaks)?

We confirm that we included all variants in the 95% credible sets in the analyses reported in Figure 4A. We also included variants in high LD ($r^2 > 0.7$ in the European population from the 1000G panels) to cover those SNPs that were absent in one or more individual studies. The method applied was previously used in similar studies (e.g. PMID: 34024118). We do agree with this Reviewer that matching only based on chromosome size and number of peaks may lead to an inflation of SNP enrichment. We therefore ran the same analysis by matching SNPs to a random pool of variants using GREGOR package (v1.4.0). The results obtained were overall very similar using both methods, and did not lead to any meaningful changes in the findings. We confirmed open chromatin regions from vascular smooth muscle cells and fibroblasts are the only ones with significant enrichment for SCAD-associated SNPs. The methods, Figure 2, Supplementary Figures 4 and 5 were updated to include the results from this method.

5. Figure 4B. What is the significance threshold? Also, when there is no symbol, is it correct to assume that there is no GTEx models? This should be clarified in the figure legend.

This is correct. The figure and legend were updated to clarify these points.

6. How were MR instruments selected? What steps did you take to ensure that the MR assumptions are respected?

In the selection process for instrumental variables (IVs) for a MR study, we ensured that the IVs meet the three assumptions recommended: 1) IVs should have a strong association with the exposure; 2) IVs should not be confounders between the exposure and outcome; and 3) IVs should only affect the outcome through the exposure. The following steps were followed to select valid IVs: 1) linkage disequilibrium (LD) clumping was applied to identify independent SNPs as candidate IVs, using a p-value threshold of $< 5 \times 10^{-8}$

⁸ and an LD $r^2 < 0.001$ within a 10000 kb window based on Europeans in the 1000 Genomes Project; 2) candidate IVs that were absent in the summary statistics data from a GWAS we used for a given outcome were excluded; 3) to minimize the risk of horizontal pleiotropy, candidate IVs that were associated with the outcome or in high to moderate LD ($r^2 > 0.6$ within a 10000 kb window) with the outcome of candidate IVs were removed.

The methods section was updated to include these details about IVs selection. (Page 14)

Reviewer #3:

Remarks to the Author:

Adlam and colleagues present the results of a genome-wide association meta-analysis of spontaneous coronary artery dissection (SCAD) totaling 1,917 SCAD cases of European ancestry and 9,292 ancestry matched controls free of SCAD from 8 prior case-control studies. A total of 17 risk loci were identified, 12 of which are novel at genome-wide significance in this study (4 of which were identified as having a suggestive level of association in the prior study of Saw et al., Nature Comm 2020). The authors confirm that SCAD is heritable and estimate that the identified loci account for ~24% of SNP-based heritability (similar to the prior estimate in Saw et al., Nature Comm 2020). The authors perform in silico based annotation of probable causal genes and identify a shared genetic basis between SCAD and arterial disease, confirming the opposite genetic effect of variants for SCAD and atherosclerotic coronary artery disease (first identified in Saw et al, Nature Comm 2020).

I complement the authors on a well-executed GWAS meta-analysis. The methods appear to be sound and conducted appropriately.

1. The identification of F3 as a risk locus is interesting and the authors propose a mechanism of decreased F3 expression as driving risk. Did the authors look at loci encoding other components of the coagulation cascade to see if there are sub genome-wide significant results in other loci? F3 forms a complex with factor VIIa, for example, so one might expect to see some signal in the F7 locus. Our current meta-analysis showed no evidence for sub-genomic association signals near *F7* and *F10*, which map in the same genomic locus on chromosome 13. We looked at tissue factor pathway inhibitor gene (*TFPI*) on chromosome 2 as well, and no association signal was detected. Future efforts including larger samples will be required to confirm the current lack of association of other partners of this pathway with the risk for SCAD.

R-Figure3. LocusZoom generated from SCAD GWAS meta-analysis results around F7/F10 on Chr13 and TFP1, on Chr2.

Similarly, did the authors consider performing a PheWAS for the F3 variant to see what other bleeding disorders (or other phenotypes) are co-associated?

We note that several SNPs from the *F3* locus were reported to associate with “end-stage coagulation” or tissue factor plasma levels in close vicinity to the association signal that we report with SCAD. However, none of these variants correlates with SCAD lead SNPs, supporting these associations to be independent from the SCAD signal. This may suggest the existence of tissue specific regulation where different variants may be involved. SCAD associated SNPs could be involved in the regulation of the expression of *F3* specifically in arteries, which is supported by GTEx data and the significant eQTL we cite in our study. This regulation could potentially be independent from different variants of *F3* expression variation that controls TF concentration in plasma.

R-Figure4: Linkage disequilibrium (r^2) in European populations (1000Genome reference panel) between SCAD associated lead variant and variants associated to Tissue Factor levels or traits related to coagulation near SCAD lead SNP.

As requested, we performed a PheWAS analysis, for rs114673, lead variant in F3 locus and its correlated SNPs ($r^2 \geq 0.80$) for traits available from UK Biobank related to bleeding or hemorrhage (list below). In addition to the absence of association with cardiometabolic traits or disease, rs114673 showed only suggestive association with plasma TF levels.

trait	beta	se	p
SCAD	0.2759	0.0474	5.82E-09
Mean platelet volume	0.006256	0.004595	1.73E-01
Platelet count	-0.002421	0.004638	6.02E-01
Platelet distribution width	-0.006149	0.00457	1.79E-01
Plateletcrit	0.000963	0.004653	8.36E-01
Antepartum haemorrhage	7.48E-05	0.0001053	4.77E-01
Arterial embolism and thrombosis	2.25E-05	9.64E-05	8.16E-01
Cause of death: gastro-intestinal haemorrhage, unspecified	0.0005683	0.0006245	3.63E-01
Cause of death: phlebitis and thrombophlebitis of other deep vessels of l	-0.0005983	0.001777	7.36E-01
Haemorrhage from respiratory passages	-0.0003364	0.000226	1.37E-01
Haemorrhage in early pregnancy	1.92E-05	0.0001358	8.88E-01
Intracerebral haemorrhage	-1.50E-05	8.54E-05	8.61E-01
Other coagulation defects	-1.50E-08	4.70E-05	1.00E+00
Other nontraumatic intracranial haemorrhage	-0.0001087	6.13E-05	7.63E-02
Other venous embolism and thrombosis	-9.80E-06	5.60E-05	8.61E-01
Phlebitis and thrombophlebitis	-5.47E-05	0.0002147	7.99E-01
Postpartum haemorrhage	-5.38E-05	8.62E-05	5.33E-01
Purpura and other haemorrhagic conditions	7.89E-05	8.74E-05	3.67E-01
Recurrent and persistent haematuria	-0.0001099	9.83E-05	2.64E-01
Self-reported brain haemorrhage	4.69E-05	5.25E-05	3.72E-01
Self-reported clotting disorder or excessive bleeding	-7.75E-06	9.59E-05	9.36E-01
Self-reported deep venous thrombosis	0.0003869	0.0004365	3.75E-01
Self-reported low platelets or platelet disorder	9.20E-05	8.89E-05	3.00E-01
Self-reported systemic lupus erythematosus or sle	-2.40E-05	0.0001025	8.15E-01
Systemic lupus erythematosus	4.75E-06	4.32E-05	9.12E-01
Unspecified haematuria	-0.0009112	0.0004546	4.50E-02
tissue_factor_level	0.0493	0.0133	2.20E-04
Ddimer_level	0.0024	0.0313	9.33E-01
coagulation_factor_measurement	-0.0176	0.0165	2.86E-01
von_Willebrand_factor_measurement	-0.0077	0.0171	6.54E-01
factor_VIII_factor_measurement	-0.0069	0.0313	8.32E-01

R-Table 3. Association between F3 lead variants and bleeding disorders phenotypes.

2. The authors suggest in a few places that this work highlights potential therapeutic strategies. Have the authors systematically characterized the loci to determine which encode druggable targets?

Our results highlight novel biological pathways based on the function of prioritized genes at risk loci, and potential therapeutic and preventive strategies based on Mendelian randomization findings supporting that controlling for blood pressure, but not LDL cholesterol would potentially benefit acute myocardial infarction presenting with SCAD. Following this reviewer's suggestion, we systematically evaluated the potential druggable targets among the genes we prioritized. We now mention this extensive analysis of druggability of gene products identified through the SCAD GWAS using the resources reported recently by Finan et al (PMID: 28356508). This analysis indicates tissue factor to be a Tier 1 druggable protein (with known bioactive drug-like small molecule binding partners and those with substantial sequence), namely target CHEMBL4081 (factor III) and CHEMBL2095194 (factor III/factor VII complex) and integrin alpha

- 1, HTRA1 and LRP1 one to be Tier 3 druggable proteins (potentially druggable targets by similarity to approved drug targets, and members of key druggable gene families). These results and corresponding methods are now included in the updated manuscript, and in Supplementary table 8.
3. Many of the major results and conclusions seem to be highlighted in prior work (with the exception of novel loci identified here) as summarized recently by Weldy et al., *Circulation Genomics and Precision Medicine* 2022. The authors might emphasize the truly novel findings here beyond what has been reported in prior publications.

Following this reviewer suggestion, we have re-ordered the results and discussion on novel loci.

4. A trivial point but the chromosome 15 designation is missing from the THSD4 locus in Table 1. We have corrected this typo in the revised Table 1.

Decision Letter, second revision:

3rd March 2023

Dear Nabila,

Your revised manuscript "Genome-wide association meta-analysis of spontaneous coronary artery dissection reveals common variants and genes related to artery integrity and tissue-mediated coagulation" (NG-A60356R2) has been seen by the original referees. As you will see from their comments below, they find that the paper has improved in revision, and therefore we will be happy in principle to publish it in *Nature Genetics* as an Article pending final revisions to comply with our editorial and formatting guidelines.

We are now performing detailed checks on your paper, and we will send you a checklist detailing our editorial and formatting requirements soon. Please do not upload the final materials or make any revisions until you receive this additional information from us.

Thank you again for your interest in *Nature Genetics*. Please do not hesitate to contact me if you have any questions.

Sincerely,
Kyle

Kyle Vogan, PhD
Senior Editor
Nature Genetics
<https://orcid.org/0000-0001-9565-9665>

Reviewer #1 (Remarks to the Author):

The authors of this manuscript describing the first large GWAS of SCAD have provided a diligent and

thorough response to my previous critique. The population is unique and multiple independent cohorts are included. I do not believe that further replication is necessary or feasible. No further comments.

Reviewer #2 (Remarks to the Author):

The authors have appropriately addressed my comments.

Reviewer #3 (Remarks to the Author):

The authors have incorporated new analyses and clarified a number of items raised in the prior round of reviews. The manuscript is improved and I have no further comments.

Final Decision Letter:

26th April 2023

Dear Nabila,

I am delighted to say that your manuscript "Genome-wide association meta-analysis of spontaneous coronary artery dissection identifies risk variants and genes related to artery integrity and tissue-mediated coagulation" has been accepted for publication in an upcoming issue of Nature Genetics.

Your paper will be published online after we receive your corrections and will appear in print in the next available issue. You can find out your date of online publication by contacting the Nature Press Office (press@nature.com) after sending your e-proof corrections. Now is the time to inform your Public Relations or Press Office about your paper, as they might be interested in promoting its publication. This will allow them time to prepare an accurate and satisfactory press release. Include your manuscript tracking number (NG-A60356R3) and the name of the journal, which they will need

when they contact our Press Office.

Before your paper is published online, we will be distributing a press release to news organizations worldwide, which may very well include details of your work. We are happy for your institution or funding agency to prepare its own press release, but it must mention the embargo date and Nature Genetics. Our Press Office may contact you closer to the time of publication, but if you or your Press Office have any enquiries in the meantime, please contact press@nature.com.

Please note that Nature Genetics is a Transformative Journal (TJ). Authors may publish their research with us through the traditional subscription access route or make their paper immediately open access through payment of an article-processing charge (APC). Authors will not be required to make a final decision about access to their article until it has been accepted. [Find out more about Transformative Journals](https://www.springernature.com/gp/open-research/transformative-journals)

Authors may need to take specific actions to achieve [compliance](https://www.springernature.com/gp/open-research/funding/policy-compliance-faqs) with funder and institutional open access mandates. If your research is supported by a funder that requires immediate open access (e.g. according to [Plan S principles](https://www.springernature.com/gp/open-research/plan-s-compliance)), then you should select the gold OA route, and we will direct you to the compliant route where possible. For authors selecting the subscription publication route, the journal's standard licensing terms will need to be accepted, including [self-archiving-and-license-to-publish](https://www.nature.com/nature-portfolio/editorial-policies/self-archiving-and-license-to-publish). Those licensing terms will supersede any other terms that the author or any third party may assert apply to any version of the manuscript.

Please note that Nature Portfolio offers an immediate open access option only for papers that were first submitted after 1 January 2021.

If you have not already done so, we invite you to upload the step-by-step protocols used in this manuscript to the Protocols Exchange, part of our on-line web resource, [natureprotocols.com](https://www.nature.com/natureprotocols). If you complete the upload by the time you receive your manuscript proofs, we can insert links in your article that lead directly to the protocol details. Your protocol will be made freely available upon publication of your paper. By participating in [natureprotocols.com](https://www.nature.com/natureprotocols), you are enabling researchers to more readily reproduce or adapt the methodology you use. [Natureprotocols.com](https://www.nature.com/natureprotocols) is fully searchable, providing your protocols and paper with increased utility and visibility. Please submit your protocol to <https://protocolexchange.researchsquare.com/>. After entering your nature.com username and password you will need to enter your manuscript number (NG-A60356R3). Further information can be found at <https://www.nature.com/nature-portfolio/editorial-policies/reporting-standards#protocols>

Sincerely,
Kyle

Kyle Vogan, PhD
Senior Editor
Nature Genetics
<https://orcid.org/0000-0001-9565-9665>